# Enhancing myocardial repair with CardioClusters

Megan M. Monsanto [1], Bingyan J. Wang [1], Zach R. Ehrenberg[1], Oscar Echeagaray[1], Kevin S. White[1], Roberto Alvarez Jr.[1], Kristina Fisher[1], Sharon Sengphanith[1], Alvin Muliono[1], Natalie A. Gude[1] & Mark A. Sussman [1✉]

Cellular therapy to treat heart failure is an ongoing focus of intense research, but progress toward structural and functional recovery remains modest. Engineered augmentation of established cellular effectors overcomes impediments to enhance reparative activity. Such 'next generation' implementation includes delivery of combinatorial cell populations exerting synergistic effects. Concurrent isolation and expansion of three distinct cardiac-derived interstitial cell types from human heart tissue, previously reported by our group, prompted design of a 3D structure that maximizes cellular interaction, allows for defined cell ratios, controls size, enables injectability, and minimizes cell loss. Herein, mesenchymal stem cells (MSCs), endothelial progenitor cells (EPCs) and c-Kit$^+$ cardiac interstitial cells (cCICs) when cultured together spontaneously form scaffold-free 3D microenvironments termed CardioClusters. scRNA-Seq profiling reveals CardioCluster expression of stem cell-relevant factors, adhesion/extracellular-matrix molecules, and cytokines, while maintaining a more native transcriptome similar to endogenous cardiac cells. CardioCluster intramyocardial delivery improves cell retention and capillary density with preservation of cardiomyocyte size and long-term cardiac function in a murine infarction model followed 20 weeks. CardioCluster utilization in this preclinical setting establish fundamental insights, laying the framework for optimization in cell-based therapeutics intended to mitigate cardiomyopathic damage.

[1] San Diego Heart Research Institute, San Diego State University, 5500 Campanile Drive, San Diego, CA 92182, USA. ✉email: heartman4ever@icloud.com

Cellular therapy continues to be pursued as an approach to treat cardiomyopathic damage following infarction in both preclinical animal models and human clinical trials[1–9]. While promising, cellular therapy has been hindered by marginal improvement in cardiac function, in part due to low engraftment and persistence of the transplanted cells[10–12]. As such, improving cell retention and survival has been a major area of research using a plethora of techniques, including biomaterials[13,14], cytokines and growth factors[15], repeated administration of cells[16], or genetic enhancement with pro-survival and anti-apoptotic genes[17]. Combinatorial cell therapies are an additional conceptual approach intended to promote additive or synergistic effects in myocardial repair[18–21]; however, cell retention and survival remain poor whether single or combinatorial cell delivery is attempted. Furthermore, coincident delivery of multiple cell populations does not ensure effective cell interaction and cross talk following injection. Therefore, strategic approaches to prolong cell retention and survival are highly sought after, preferably also favoring cell phenotypic characteristics that promote short-term mitigation of injury and long-term recovery of myocardial structure and function.

Traditional monolayer culture, while both prevalent and convenient, promotes loss of cell identity and spatial organization, thereby impacting many cellular aspects, including cell morphology, proliferation, differentiation, viability, and transcriptome profile[22–24]. Approaches intended to mitigate deleterious effects of artificial in vitro environments through recreating native 3D architecture include engineered heart tissue grafts[25,26], organoids[27,28], and 3D bioprinting[29,30]. When grown in 3D cells aggregate and form complex cell-cell and cell-matrix interactions that mimic the natural environment found in vivo, allowing for cell differentiation and tissue organization not possible in conventional 2D culture systems[31]. For example, cardiosphere formation enhances cellular communication through gap junction protein connexin 43 fostering signal propagation and cell differentiation[32–34]. However, cardiospheres are formed by random cell aggregation leading to inconsistent cell characterization markers[35,36], variable sphere size, and muddled communication within the aggregate microenvironment[9,35,36]. A preferable methodology would utilize combinations of defined cell populations aggregated in 3D under controlled conditions that allow for intact delivery without dissociation of the spherical microenvironment to promote preservation of cellular phenotype and optimize cellular retention. This combination of desired features formed the rational basis for our approach distinct in several critical aspects from cardiosphere-derived cell therapy.

A "next generation" concept for a cell therapeutic based upon the aforementioned literature precedents would involve (1) combinations of multiple cardiac-derived cell types, (2) ex vivo culture adaptation to a 3D microenvironment to maximize cellular interaction, (3) control over cell composition, cell number, and aggregate cell volume, and (4) capability to deliver ex vivo formed cell aggregate niches intact to facilitate preservation of microenvironment biological properties and retention. Guided by these stipulated parameters to create an artificially engineered cell aggregate, inception began with essential methodological studies to isolate and expand three distinct cardiac-derived interstitial cell types from a single human heart biopsy as previously reported by our group[37]. Human cardiac tissue of varying age and gender was used as source material to isolate c-Kit$^+$ cardiac interstitial cells (cCIC), mesenchymal stem cells (MSCs), and endothelial progenitor cells (EPCs). Having isolated and characterized three distinct cardiac interstitial cell types each known to possess beneficial properties to blunt cardiomyopathic damage, a protocol was designed for spontaneous self-assembly of mixed cell populations into an optimal conceptual engineered solution termed a CardioCluster.

CardioClusters harness the distinct phenotypic attributes of three well defined cardiac cell populations. MSCs support myocardial reparative activity through secretion of paracrine factors that activate endogenous cells, promote angiogenesis, protect cardiomyocytes, and reduce scar formation[38,39]. MSCs also secrete cell adhesion molecules such as integrins and cadherins integral to cellular aggregation[40]. EPCs promote paracrine dependent vasculogenesis and angiogenesis and differentiate into mature endothelial cells[41]. Prior studies demonstrate EPCs transplanted in vivo are capable of forming microvessels, but regress without MSCs to support vessel maturity[42,43]. cCICs distributed in the CardioCluster contribute to support of myocardial homeostasis, response to injury, and remodeling. CardioCluster characteristics are particularly well suited to mediating myocardial repair in the wake of acute pathological damage by providing a more natural niche microenvironment for augmented delivery and extended functional activity to improve the outcome of cellular therapy.

CardioClusters advance the application of combinatorial cell therapy by integrating complementary and synergistic properties from multiple cardiac-resident cell types into a single injectable product. CardioCluster formation is a rapid, reproducible, and controllable process demonstrated in preclinical testing to mediate significant improvements in myocardial structure and function following infarction injury.

## Results

**Characterization of three distinct cardiac cell populations.** CardioClusters are comprised of three cardiac-resident cell populations concurrently isolated as previously described by our group with full phenotypic characterization[37]. Cell surface marker profiling of cell lines used to produce CardioClusters were similar to previous findings (cCIC express c-Kit$^{high}$, CD90$^{high}$, CD105$^{low}$, CD133$^{low}$, CD45$^{neg}$; EPC express CD133$^{high}$, CD105$^{high}$, c-Kit$^{low}$, CD90$^{neg}$, CD45$^{neg}$; and MSC express CD90$^{high}$, CD105$^{high}$, c-Kit$^{low}$, CD133$^{low}$, CD45$^{neg}$)[37]. Commitment toward angiogenic and smooth muscle fates was assessed for the three cardiac-derived cell populations relative to control cell lines human umbilical vein endothelial cells (HUVECs) and bone marrow-derived MSCs (BM MSCs). Tube formation assays demonstrated robust angiogenic responses from both EPCs and HUVECs in vitro using growth factor reduced Matrigel (Supplementary Fig. 1a). Transcript levels of endothelial-related genes *CD31* and *von Willebrand factor (vWF)* were elevated in HUVECs and EPCs ($p < 0.001$ versus cCICs; Supplementary Figs. 1b and 3c). Smooth muscle actin (*SMA*) transcripts were highly expressed in BM MSCs and cardiac MSCs ($p < 0.001$ versus cCICs), with both endothelial populations (EPC or HUVEC) expressing near undetectable levels ($p < 0.05$ versus cCICs; Supplementary Fig. 1d). *GATA4* was expressed by cCICs ($1.0 \pm 0.05$) and to a lesser extent by EPCs ($0.87 \pm 0.03$) and MSCs ($0.33 \pm 0.01$), with non-cardiac controls expressing undetectable levels (Supplementary Fig. 1e). Collectively, these three cardiac-derived cell populations recapitulate and validate previous results of phenotypic characterization for cell types obtained using our published protocol[37]. Distinct phenotypic properties of these three cardiac-derived cell populations fulfills the conceptual design of combining multiple cell types for CardioClusters formation.

The three cell populations were modified with lentiviral vectors to introduce fluorescent proteins for tracking purposes (eGFP tagged cCICs [green], mOrange tagged EPCs [blue], and Neptune tagged MSCs [red]; tagging efficiency $99.1 \pm 0.2\%$; Supplementary

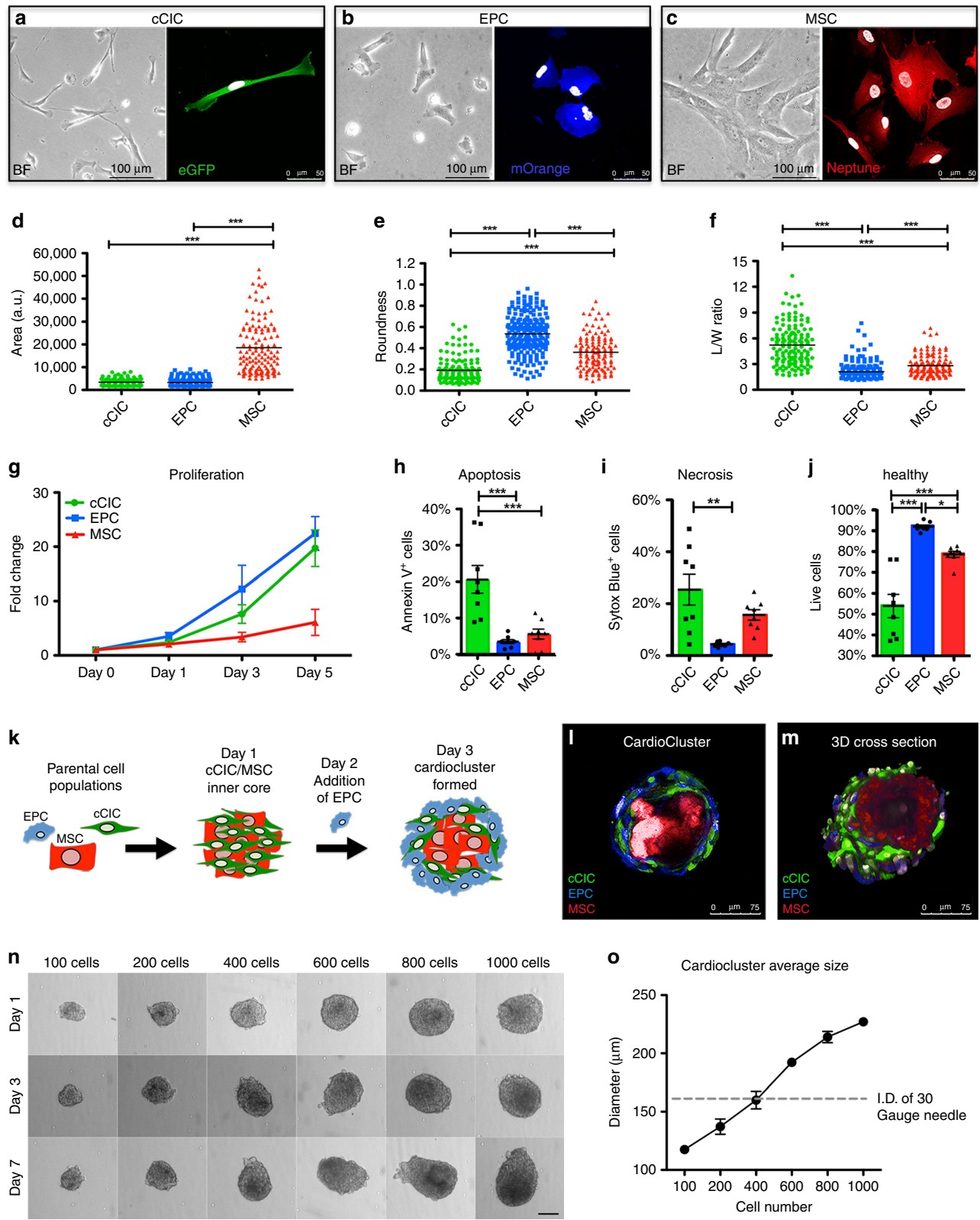

Fig. 2a, b). Distinct morphology for each cell population is evident in representative brightfield images with companion immunofluorescent images demonstrating corresponding fluorophore expression in cCICs (Fig. 1a), EPCs (Fig. 1b), and MSCs (Fig. 1c). Cell morphology measurement of area, roundness, and L/W ratio for each cell type confirmed distinct phenotypes (Fig. 1d–f). MSCs were significantly larger (18,563 ± 1,021)

relative to both cCIC (3383 ± 121) and EPC (3272 ± 102) (Fig. 1d). EPCs were significantly rounder (EPC, 0.55 ± 0.012; cCIC, 0.19 ± 0.0097; MSC, 0.36 ± 0.015) (Fig. 1e), while cCICs show increased L/W ratio (cCIC, 5.2 ± 0.19; EPC, 2.1 ± 0.063; MSC, 2.8 ± 0.11) (Fig. 1f). Morphometric parameters clustered by cell type (Supplementary Fig. 3), with minor variation between heart samples. EPCs exhibited a proliferative rate similar to cCICs, with

**Fig. 1 Three distinct cardiac cell lineages generate CardioClusters. a–c** Representative brightfield (BF) and immunofluorescent images for cCIC (eGFP+) (**a**), EPC (mOrange+) (**b**) and MSC (Neptune+) (**c**). Scale bars: brightfield, 100 μm; immunofluorescent, 50 μm. DAPI to visualize nuclei (white). **d–f** Cell morphometric parameters measuring area (a.u. arbitrary units; **d**), roundness (**e**), and length-to-width (L/W) ratio (**f**). Data in **d**, **e** represent mean ($n = 3$ independent human heart isolations; minimum of 30 cells traced per cell type, per heart) ± SEM. Data are presented as one-way ANOVA with Bonferroni comparison of all pairs, ***$p < 0.0001$. **g** Cardiac cell proliferation measured using CyQuant assays at day 0, day 1, day 3, and day 5. Data represent mean ($n = 3$ independent heart samples, run in quadruplicate) ± SEM. **h–j** Annexin V/Sytox Blue labeling for apoptotic (**h**), necrotic (**i**) and healthy (**j**) cells following cell death assay: 24 h of low serum (75% serum reduction) followed by 4 h treatment with 30 μM $H_2O_2$ in low serum medium. Data represent mean ($n = 4$ independent heart samples, each experiment run in duplicate) ± SEM. Data are presented as 1-way ANOVA with Tukey's multiple comparisons test, (**h**) ***$p < 0.001$, (**i**) **$p = 0.001$, (**j**) *$p = 0.028$ and ***$p < 0.001$. **k** Schematic showing CardioCluster formation using three cell populations isolated from human heart tissue: MSC (red), cCIC (green), EPC (blue). **l** Live CardioCluster visualized by endogenous fluorescent tags showing cCIC (eGFP; FITC channel; green), MSC (Neptune; APC channel; red) and EPC (mOrange; PE channel; blue). Scale bar, 75 μm. **m** 3D cross section of a CardioCluster lightly fixed in 4% paraformaldehyde and stained with 4′,6-diamidino-2-phenylindole (DAPI; white) to visualize nuclei and cells exhibiting fluorescent tags without the need for antibody labeling. Scale bar, 75 μm. **n** Representative brightfield images of CardioClusters ranging in size from 100 to 1000 cells cultured over a 7-day period. Scale bar, 100 μm. **o** Determination of cell number enabling CardioClusters to pass through a 30-gauge needle, which has an inner diameter (I.D.) of 159 μm. Graph plot of CardioCluster diameters averaged over 3 days ($n = 4$ (100 cell), $n = 6$ (200 cell), $n = 6$ (400 cell), $n = 7$ (600 cell), $n = 5$ (800 cell), $n = 5$ (1000 cell) CardioClusters per cell number group). Source data are provided as a Source data file.

both populations showing increased proliferation over MSCs based on CyQuant proliferation assays (Fig. 1g). EPCs were significantly more resistant to cell death and retained 92 ± 0.76% cell viability, versus only 54 ± 5.6% for cCIC and 79 ± 1.5% for MSCs after 4 h $H_2O_2$ treatment (Fig. 1h–j). Cumulatively, characterization showed phenotypic and biological distinctions between cardiac interstitial cell populations fundamental to CardioCluster design and utility, such as elevated resistance to oxidative stress-induced cell death, high proliferative activity, and pro-angiogenic nature of EPCs.

**Generation of CardioClusters**. CardioClusters are formed in a two-step process (Fig. 1k). cCICs and MSCs are seeded to form the inner core, with EPCs added 24 h later to provide an endothelial cell-enriched outer layer for the CardioCluster. The outer EPC layer provides enhanced resistance to oxidative stress relative to the more sensitive cCICs and MSCs within the CardioCluster core (Fig. 1j). Individual cells within an assembled CardioCluster were readily visualized with their cognate fluorophore tags, obviating the need for antibody-mediated detection (Fig. 1l, m). CardioCluster size reproducibly and predictably corresponds to cell number seeded per microwell (Fig. 1n). CardioCluster ranging from 100–1000 cells were examined to determine changes in size and morphology over a 7-day time period (Supplementary Fig. 2d). CardioCluster diameter and area increased over 7 days, except for CardioClusters seeded with 1000 cells, whose diameter plateaued after day 3. This finding, also previously observed with 3D aggregated cells[44], is consistent with reduced oxygen and nutrient diffusion within dense cellular structures >200 μm[45,46].

Preserving CardioCluster 3D structural integrity for intramyocardial delivery to promote intercellular contact and enhance retention is essential to improve upon typical approaches involving dissociated single cell suspensions. Intramyocardial injection for cell delivery in murine hearts uses a standard 30-gauge needle with a 159 μm internal diameter ([I.D.]), so CardioClusters were engineered for a diameter allowing for injection to preserve 3D structure. The maximum number of cells that could comprise a CardioCluster and pass through a 30-gauge needle is 400 cells based on morphometric quantitative analysis (Fig. 1o). EPC, cCICs and MSCs were combined in a 3:2:1 ratio, based upon the consideration of larger MSC size occupying relatively more volume relative to cCIC or EPC (Fig. 1d). CardioCluster spontaneous self-assembly as revealed using time-lapse video microscopy shows the MSC population immediately migrating towards the central core (Supplementary Video 1). cCIC/MSC interaction was allowed to progress for 24 h, at which point EPCs were added to interact with established cCIC/MSC

cores. Interestingly, EPCs initially form their own clusters instead of adhering to cCIC/MSC cores and then subsequently envelop the cCIC/MSC core (Supplementary Video 1 and Supplementary Fig. 2). Architecture of an MSC-enriched core was invariant regardless of seeding sequence, as plating of cCIC+EPCs prior to adding MSCs consistently resulted in MSCs migrating and localizing within the CardioCluster core rather than surface (Supplementary Video 2) consistent with the preferential localization of MSC to hypoxic environments[47,48]. CardioCluster formation consistently occurs with MSCs in the core and cCIC/EPCs on the outer layers.

CardioClusters possess a high percentage of live cells maintained within the 3D structure (93.9–98% of cells alive; Supplementary Fig. 4). Robust vitality of CardioClusters was confirmed by recovery from long-term liquid nitrogen storage, where the percentage of live cells was comparable to that of control non-frozen CardioClusters (Supplementary Fig. 4a, b). When cultured on standard tissue culture-treated plastic, cells adhered and migrated out from the CardioCluster whether frozen or non-frozen, with comparable cell morphology (Supplementary Fig. 4c). These findings support "off-the-shelf" feasibility of using frozen banked CardioClusters for therapeutic purposes rather than necessitating de novo creation prior to use.

**CardioClusters exert protective effects in vitro**. Previous studies have found protective effects mediated by cCICs and MSCs conferred upon serum starved neonatal rat cardiomyocytes (NRCMs) in co-culture[49]. To confirm the effect of cell rescue, the beneficial effects mediated by CardioClusters were assessed by co-culture with NRCMs in serum depleted conditions relative to effects conferred by cCIC, EPCs, MSCs, and a combined mixture of cCIC + EPC + MSC (C + E + M) (Fig. 2a). NRCMs maintained in low serum (0.5%) were smaller relative to NRCMs maintained in high serum condition (10%) (Fig. 2b, c, Supplementary Fig. 5). CardioCluster co-culture with low serum treated NRCMs restored cardiomyocyte size within 24 h relative to all other treatments (p < 0.05; Fig. 2b, c, Supplementary Fig. 5) and also increased mRNA expression for *Desmin*, a muscle-specific type III intermediate filament protein ($p < 0.01$; Fig. 2d). Furthermore, CardioCluster co-culture increased mRNA for *Sdf-1* ($p < 0.05$; Fig. 2e), a cardioprotective cytokine and chemotactic factor for MSCs that plays an additional role in recruitment of EPCs important for angiogenesis[50,51]. Importantly, CardioClusters offered significantly greater protection upon NRCM than actions exerted by any individual parental population (cCICs, EPCs, MSCs) or the combined C + E + M mixed population. Collectively, this in vitro study demonstrates superior

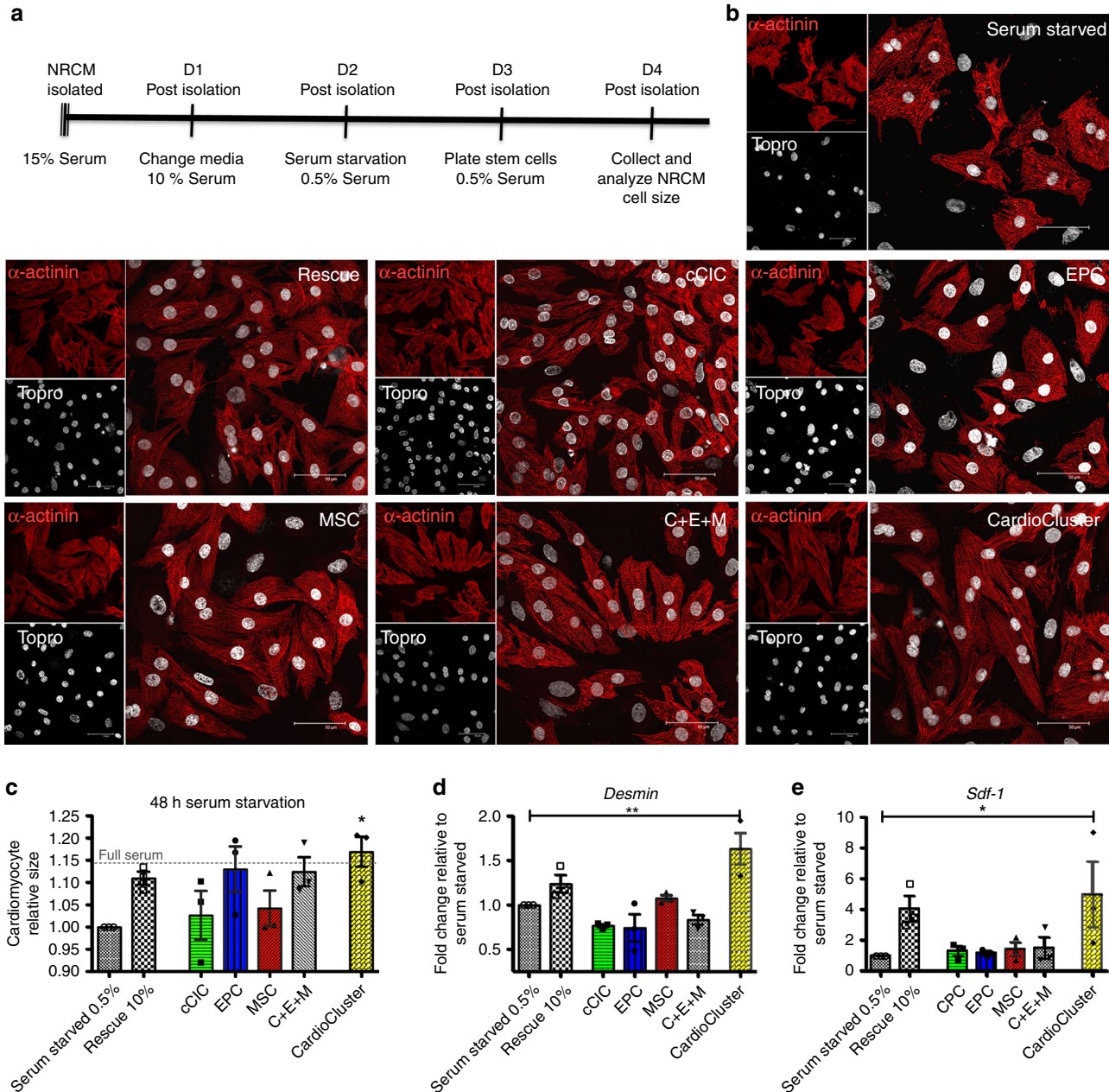

**Fig. 2 CardioClusters protect cardiomyocytes against low serum conditions in vitro. a** Timeline for neonatal rat cardiomyocyte (NRCM) low serum assay over a 4-day (D) period. **b** Representative images of NRCM conditions: serum starved (48 h of 0.5% serum), rescued (24 h of 0.5% serum with 10% serum added for additional 24 h), and experimental groups (24 h of 0.5% serum with addition of either cCIC, EPC, MSC, C + E + M, or CardioCluster for additional 24 h). Cardiomyocytes visualized by staining with sarcomeric actinin (α-actinin; red). TO-PRO-3 iodide (Topro; white) used to visualize nuclei. Scale bar, 50 μm. **c** Quantitation of cardiomyocyte size relative to serum starved control. **d, e** Gene expression of *desmin* (**d**) and *sdf-1* (**e**) in cardiomyocytes with and without the addition of cells. Data in **c–e** represent mean ($n = 3$ independent NRCM preparations) ± SEM. Data are presented as one-way ANOVA with Dunnett's comparison test, **c** *$p = 0.028$, **d** **$p = 0.003$, **e** *$p = 0.038$, versus serum starved. Source data are provided as a Source data file.

protective effects of CardioClusters for NRCMs in response to serum starvation challenge.

**Paracrine expression increased in CardioClusters after co-culture.** Paracrine factor action is considered a primary mechanism for cardioprotection[52], accordingly mRNA transcript level expression for growth and immunomodulatory factors was assessed after CardioCluster co-culture for 5 days with serum depleted NRCM (Supplementary Fig. 6a). mRNA levels for CardioClusters, parental cells, and the C + E + M mixed population were measured by separating fluorescently tagged cells away from

the NRCM population using flow cytometric sorting (Supplementary Fig. 5). mRNA transcript levels for *insulin-like growth factor (IGF)* and *interleukin-6 (IL-6)* were elevated in CardioClusters co-cultured with NRCMS relative to any of the individual parental population (cCICs, EPCs, MSCs) or the combined C + E + M mixed population ($p < 0.001$ and p < 0.05 respectively, versus cCIC; Supplementary Fig. 6b, c). *IGF* exerts chemotactic and growth-stimulatory effects[52] in addition to anti-apoptotic properties[53–55]. Early release of anti-inflammatory cytokines such as *IL-6* after acute cardiac damage has been shown to be beneficial by signaling protective responses in local tissue and initiating

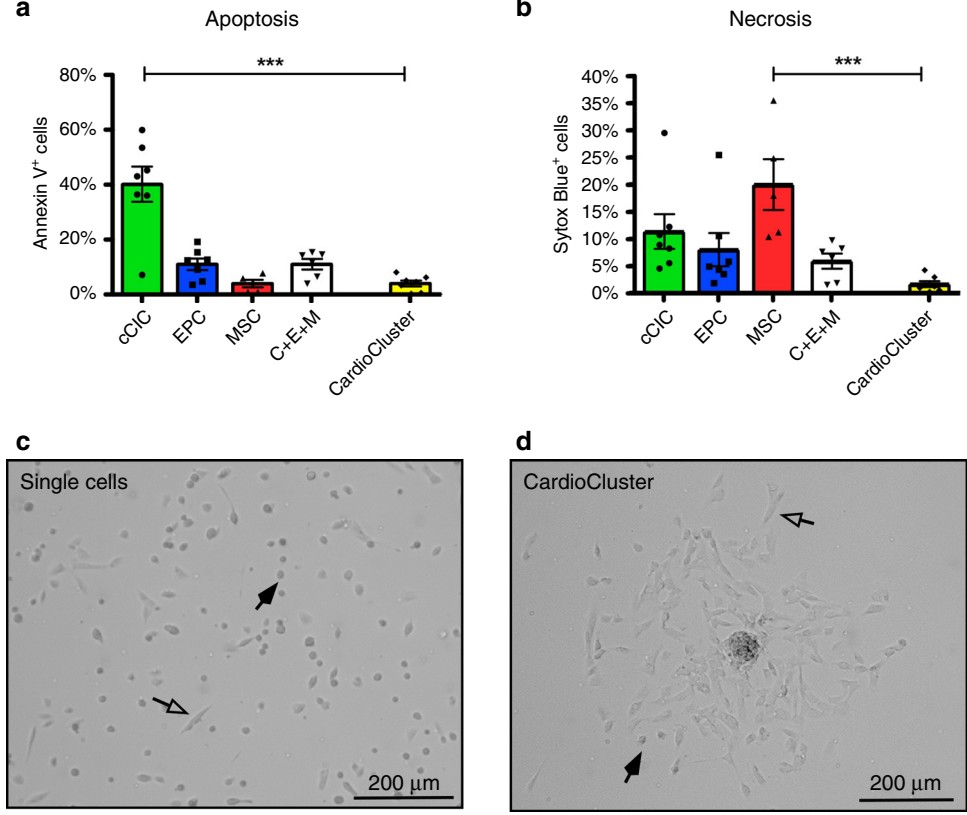

**Fig. 3 CardioClusters are protective against oxidative damage. a–d** Cell death assay performed on cardiac cell populations under 24 h of low serum (75% serum reduction), followed by 4 h of treatment with 30 μM $H_2O_2$ in low serum medium. **a, b** Apoptosis visualized by Annexin V (**a**) and necrosis visualized by Sytox Blue (**b**). Data in **a** and **b** represent mean ($n = 4$ cCIC; $n = 4$ EPC; $n = 3$ MSC; $n = 3$ C + E + M; $n = 4$ CardioCluster, independent experiments run individually or in duplicate) ± SEM. Data are presented as one-way ANOVA with Dunnett's comparison test, **a** ***$p < 0.001$, **b** ***$p < 0.001$, versus CardioCluster. **c, d** Brightfield images of single cells (**c**) and CardioClusters (**d**) 4 h after $H_2O_2$ treatment. Transparent arrows highlight adherent (healthy) cells and black arrows highlight cells that have rounded up and detached from the tissue culture surface, most likely undergoing cell death. Source data are provided as a Source data file.

wound healing[56]. Additionally, the cardioprotective cytokines *SDF-1* and *hepatocyte growth factor* (*HGF*), both trended towards increased expression in CardioClusters following co-culture experiments (Supplementary Fig. 6d, e). *HGF* stimulates cell proliferation, motility, morphogenesis, angiogenesis and importantly tissue regeneration[55,57]. Collectively these results show that at the transcript level CardioClusters induction of paracrine factors *IGF* and *IL-6* exceeds that of parental cell populations or C + E + M group when co-cultured with serum depleted NRCMs, suggesting that paracrine factor release may be responsible for the protective effects observed with co-culture (Fig. 2).

To further investigate the role of the individual cell types, several mRNAs associated with lineage specification were analyzed following co-culture of CardioClusters or parental cell populations with NRCMs. *GATA4* showed the highest expression in cCIC co-culture. Predictably, EPCs displayed the largest induction of endothelial marker *CD31*, whereas MSCs induced *SMA* gene expression after 5 days of co-culture with NRCMs (Supplementary Fig. 6f–h). Neither *CD31* nor *SMA* were significantly upregulated in CardioCluster group (Supplementary Fig. 6g, h).

**CardioClusters are resistant to oxidative stress.** CardioClusters were substantially more resistant to cell death induced by 4 h of $H_2O_2$ treatment after overnight low serum culture (Fig. 3). Dying cells were divided into groups of early apoptosis, late apoptosis, and necrosis based on Annexin V and Sytox Blue staining

(Supplementary Fig. 7). CardioClusters showed significantly fewer dying cells relative to cCICs and MSCs ($p < 0.001$ [apoptosis] and $p < 0.001$ [necrosis] respectively; Fig. 3b and Supplementary Fig. 7f) as visually evidenced by fewer cells rounding up and detaching from the tissue culture dish (transparent arrows; Fig. 3c, d). These results demonstrate superiority of CardioClusters to survive oxidative stress challenges in vitro and their potential as a therapeutic.

**CardioClusters improve myocardial parameters following injury.** Therapeutic efficacy of CardioClusters was assessed in a murine experimental myocardial infarction injury model of permanent coronary artery occlusion. Xenogenic human cell treatment into NOD^SCID recipient mice was performed at the time of infarction with direct comparison between CardioClusters and the C + E + M combined population group administered as a single cell suspension mixture. Myocardial structure and function were assessed by parasternal long axis echocardiography for four experimental groups: non-injured sham, CardioCluster, C + E + M, and vehicle-treated (Fig. 4a). All groups had comparable reduction in cardiac function at 1 week post injection (wpi) demonstrating consistency of infarction injury (Fig. 4b–d, f, g and Supplementary Data 3) with average EF for all infarcted groups of approximately 30% (CardioCluster, 27 ± 2.9%; C + E + M, 32 ± 2.2%; Vehicle, 29 ± 2.0%; Supplementary Fig. 8). The CardioCluster-treated group showed significant cardiac functional improvement starting 4 wpi, which was sustained

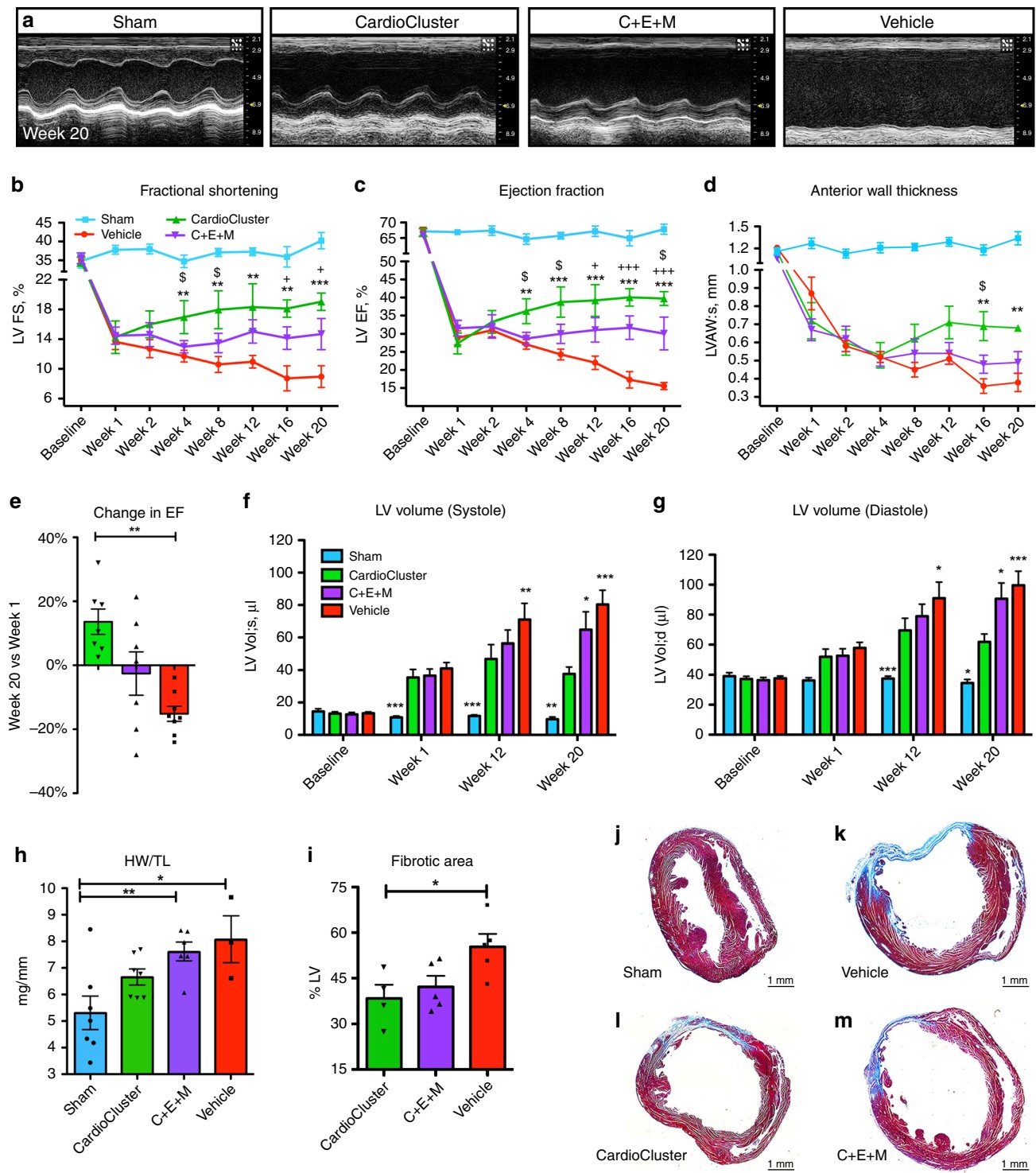

throughout the 20-week time course, with increased fractional shortening (FS; Fig. 4b) and ejection fraction (EF; Fig. 4c) versus C + E + M treatment 4 and 8 wpi and was significant at study completion for EF. In comparison, EF and FS improvements in the C + E + M treated group only began to appear at 12 and 16 wpi, respectively (Fig. 4b, c). Terminal EF measurements at 20 wpi show EF value is highest in the CardioCluster group relative to vehicle only or C + E + M groups (40 ± 1.9% versus 16 ± 1.0% or 30 ± 4.5%, respectively; Supplementary Fig. 8b–d). CardioCluster treatment shows a 45 ± 7% improvement in EF at study completion relative to the starting value at week 1.

In contrast, EF at study completion relative to week 1 for the C + E + M or vehicle only-treated groups decreased by 5 ± 14% and 46 ± 3%, respectively (Fig. 4e). Furthermore, CardioCluster treatment group exhibits significantly smaller left ventricular internal diameter both in systole (LVID;s) and diastole (LVID;d), as well as reduction in LV end systolic and diastolic volumes (LV Vol;s and LV Vol;d). Heart rate was not significantly different among treatment groups (Supplementary Fig. 9c). Structural and functional data are detailed in Supplementary Data 3.

CardioCluster superiority for restoring myocardial structure and function relative to the mixed population C + E + M is

**Fig. 4 CardioCluster treatment improves LV wall structure and cardiac function after myocardial injury. a** Representative 2D echocardiography images (M-mode) at study completion (week 20). Para-sternal short-axis view showing LV anterior wall and posterior wall movement. **b–d** Longitudinal assessment of LV fractional shortening (FS, %) (**b**), ejection fraction (EF, %) (**c**), and anterior wall thickness in systole (LVAW;s, mm) (**d**) over 20 weeks. Data are presented as one-way ANOVA with Newman-Keuls comparison test, analyzed per timepoint, **$p < 0.01$, ***$p < 0.001$, CardioCluster versus vehicle. +$p < 0.05$, +++$p < 0.001$, C + E + M versus vehicle. $p < 0.05$, CardioCluster versus C + E + M. Data represents mean (number of mice specified in Supplementary Data 3) ± SEM. **e** Change in EF from week 1 to week 20. Data represents mean ($n = 7$ CardioCluster-treated mice; $n = 7$ C + E + M-treated mice; $n = 8$ vehicle-treated mice) ± SEM. Data are presented as one-way ANOVA with Dunnett's comparison test, **$p = 0.003$, versus CardioCluster. **f, g** Bar graph showing LV volume in systole (Vol;s, μl; **f**) and in diastole (Vol;d, μl; **g**). Data represents mean (number of mice specified in Supplementary Data 3) ± SEM. Data are presented as two-way ANOVA with Dunnett's comparisons test, **f** ***$p < 0.001$, week 1; **$p = 0.002$, ***$p < 0.001$, week 12; *$p = 0.015$, **$p = 0.009$, ***$p < 0.001$, week 20 and **g** *$p = 0.011$, ***$p < 0.001$, week 12; *$p = 0.012$ [C + E + M], *$p = 0.015$ [sham], ***$p < 0.001$, week 20; versus CardioCluster. **h** Heart weight to tibia length ratio (HW/TL; mg/mm) at week 20. Data represents mean (n number of mice as indicated) ± SEM. Data are presented as one-way ANOVA with Dunnett's comparison test, *$p = 0.012$, **$p = 0.009$, versus sham. **i–m** Masson's Trichrome staining used to evaluate LV fibrotic area. Percentage of fibrotic LV in CardioCluster, C + E + M and vehicle-treated hearts (**i**). Data represents mean (n number of mice as indicated) ± SEM. Data are presented as one-way ANOVA with Dunnett's comparison test, *$p = 0.028$, versus CardioCluster. Representative histology sections of sham (**j**), vehicle (**k**), CardioCluster (**l**), and C + E + M (**m**) treated hearts 20 weeks after MI. Collagen-rich areas (scar tissue) are colored in blue and healthy myocardium in red. Images are representatives from the indicated mice in (**i**). Scale bars, 1 mm. Source data are provided as a Source data file.

further reinforced by tissue morphometry and hemodynamic measurements. Cardiac hypertrophy was not a contributing factor to increasing anterior wall thickness (AWT) at 20 weeks in the CardioCluster-treated group (Fig. 4d) as heart weight to tibia length ratios did not increase (HW/TL; Fig. 4h) relative to the sham-operated control. In contrast, a significant increase in HW/TL is present in both vehicle control as well as C + E + M treatment groups. Fibrotic area is significantly smaller in CardioCluster-treated mice compared to vehicle at 20 wpi (38.4 ± 4.5% of CardioCluster LV versus 55.3 ± 4.3% of vehicle LV; Fig. 4i–m) although infarct size is not significantly different between hearts receiving C + E + M or CardioClusters. Invasive hemodynamic measurement validates functional superiority of the CardioCluster-treated group showing significantly improved developed pressure over time (dP/dT) versus vehicle (Supplementary Fig. 9d), in addition to increasing left ventricular developed pressure (LVDP) and $P_{max} - P_{min}$ (Supplementary Fig. 9e). Collectively, these findings are evidence that CardioClusters offer significantly greater benefit for restoration of myocardial performance in this murine myocardial infarction injury model.

Speckle-tracking based strain analysis is a highly sensitive echocardiographic technique for assessing left ventricular (LV) function[58,59]. LV function was similarly reduced in all infarcted mice at 1 wpi (Fig. 5a). Progressive changes consistent with adverse ventricular remodeling occur in vehicle-treated animals in agreement with conventional echocardiographic measures of function (Fig. 4). LV systolic deformation in the CardioCluster-treated group showed significant improvement starting at 8 wpi and progressing through 20 wpi compared to vehicle-treated animals (Fig. 5b–d). Radial strain measurements at 8 and 20 wpi confirmed significant functional benefit provided by CardioCluster treatment versus 2D cultured parental C + E + M mixed population ($p < 0.05$ and $p < 0.01$ respectively; Fig. 5c). Peak longitudinal strain was improved in the C + E + M treated group versus vehicle by week 20 ($p < 0.05$; Fig. 5d). Regional strain measurements assessing the area of injury further demonstrate significant improvement in LV function for CardioCluster-treated animals (Fig. 5e–h). In the area of injury, the absolute difference in radial strain (week 1-to-week 20) for CardioCluster-treated group was 7.59 ± 1.28% which was significantly improved relative to C + E + M and vehicle-treated groups (−1.56 ± 1.96% and −1.88 ± 1.61%, respectively, $p < 0.01$ and $p < 0.001$ versus CardioCluster group, Fig. 5g). Similar improvement was seen for absolute difference in longitudinal strain 20 wpi.

**CardioClusters engraft and persist post intramyocardial injection.** Characteristics of CardioClusters including multicellular 3D architecture and enhanced survival are attractive features to mediate increased persistence following delivery compared to dissociated single cell suspensions such as the C + E + M mixed population. CardioCluster persistence in vivo was longitudinally assessed over a 20-week period by confocal microscopy (Fig. 6). CardioCluster localization was tracked with co-injection of FluoSpheres tracking beads in pilot studies to confirm delivery location in tissue sections (Fig. 6a–d). Cryosectioned hearts allowed for direct visualization of fluorophore tags without antibody labeling. All three constituent cell types were readily visualized in the myocardial wall, with MSCs at the center of the CardioCluster surrounded by a layer of cCICs and EPCs, similar to architecture observed in vitro (Fig. 1). With CardioCluster localization confirmed coincident with the injection site, subsequent injections and imaging were performed without FluoSpheres tracking beads for long-term functional studies. CardioClusters were clearly visible within the myocardium at serial time points: 1 day, 3 days, 1-, 4-, 12-, and 20-weeks post injection (Fig. 6e–k). Antibody labeling confirmed CardioCluster persistence up to 20 weeks post injection (Fig. 6h–k).

**CardioClusters increase capillary density in infarct zone.** Capillary density was measured in the infarct, border zone, and remote regions at 20 wpi. Non-injured controls (sham) serve as the control group compared to injured hearts (Fig. 7). Notably, the CardioCluster group exhibited significantly more isolectin labeled vessels in the infarct region at 20 wpi versus both vehicle and C + E + M-treated groups ($p < 0.001$, Fig. 7a, e–g). CardioCluster group capillary density in the infarct region increased 62% or 83% versus the C + E + M or vehicle only control, respectively. Within the infarct border zone at 20 wpi, both CardioCluster and C + E + M-treated groups trended toward increased capillary density versus vehicle control, but not achieving significance (Fig. 7b). The remote region did not significantly increase capillary density in either CardioCluster or C + E + M groups relative to vehicle or sham (Fig. 7c, d). Taken together, these data demonstrate a superior level of microvascularization prompted by CardioCluster treatment relative to dissociated mixed cell preparation or vehicle only control groups.

**CardioCluster treatment preserves cardiomyocyte size.** Cell therapy reduces hypertrophic remodeling following pathologic

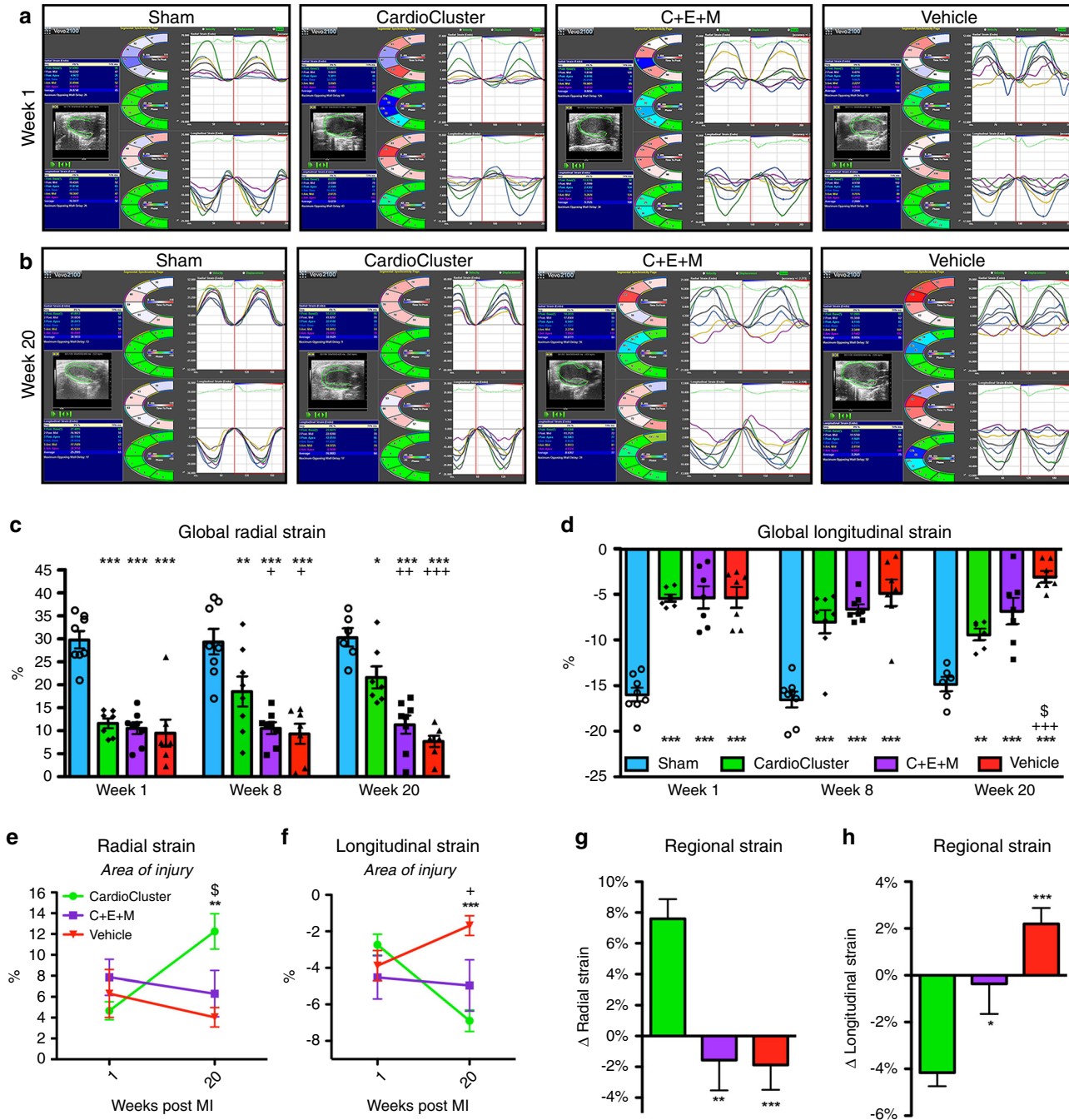

**Fig. 5 Impact of CardioClusters on cardiac function measured by cardiac strain. a**, **b** Representative images of long-axis echocardiography recording (left panel), with cross-sectional segment synchronicity map (middle panels), and radial and longitudinal strain curves (right panels) at week 1 (**a**) and week 20 (**b**). Strain curves (representing strain measures over time) are generated for the 6 standard myocardial regions, with a 7th line (shown in black) denoting the average strain. **c**, **d** Speckle tracking echocardiography analysis used to determine global peak radial strain (%) (**c**) and global peak longitudinal strain (%) (**d**) at week 1, week 8, and week 20. Data represents mean (n number of mice as indicated) ± SEM. Data are presented as two-way ANOVA with Tukey's multiple comparisons test, (**c**) ***$p < 0.001$, versus sham, week 1; **$p = 0.003$, ***$p < 0.001$, versus sham, +$p = 0.041(C + E + M)$, +$p = 0.018$ (vehicle), versus CardioCluster, week 8; *$p = 0.048$, ***$p < 0.001$, versus sham, ++$p = 0.007$, +++$p < 0.001$, versus CardioCluster, week 20. **d** ***$p < 0.001$, versus sham, week 1; ***$p < 0.001$, versus sham, week 8; **$p = 0.004$, ***$p < 0.001$, versus sham, +++$p < 0.001$, versus CardioCluster, $$p = 0.042$, versus $C + E + M$, week 20. **e**, **f** Speckle tracking echocardiography analysis used to determine regional peak radial and longitudinal strain (%) in the area of injury at week 1 and week 20. Data are presented as 2-way ANOVA with Tukey's comparison tests, **e** $$p = 0.031$, CardioCluster versus $C + E + M$, **$p = 0.001$, CardioCluster versus vehicle, **f** ***$p < 0.001$, CardioCluster versus vehicle, +$p = 0.020$, $C + E + M$ versus vehicle. **g**, **h** Total change in regional radial and longitudinal strain from week 1 to week 20 in the area of injury. Data are presented as 1-way ANOVA with Dunnett's comparison test, **g** **$p = 0.002$, ***$p < 0.001$, **h** *$p = 0.013$, ***$p < 0.001$, versus CardioCluster. **e–h** Data represents mean ($n = 9$ CardioCluster-treated mice [week 1] and 7 CardioCluster-treated mice [week 20]; $n = 6 C + E + M$-treated mice; $n = 7$ vehicle-treated mice [week 1] and 8 vehicle-treated mice [week 20]) ± SEM. Source data are provided as a Source data file.

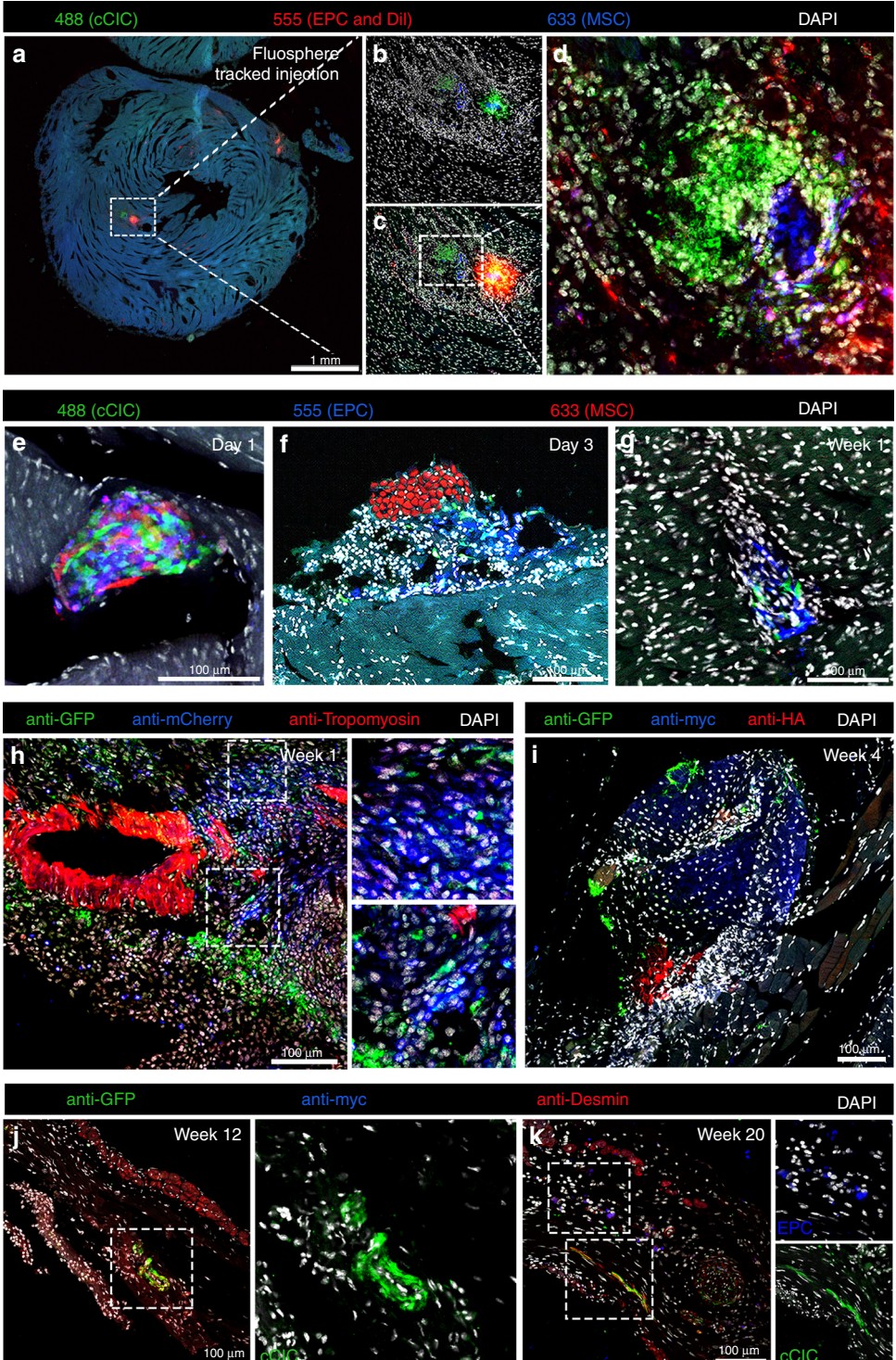

**Fig. 6 CardioCluster show enhanced engraftment and persistence in the myocardial wall. a** Immunofluorescent tile scan of a cryosectioned heart from an uninjured animal injected with CardioClusters day 3 post-injection tracked by FluoSpheres with no antibody labeling required for visualization of cells. **b–d** Higher magnification of areas within white dotted boxes. **b** Illustrates the removal of the 555 nm channel in order to better visualize cCIC (green) and MSC (blue), without FluoSpheres. **c, d** In all, 555 nm channel restored (**c**) and field of view magnified (**d**). **e–g** Immunofluorescent images from cryosectioned MI hearts injected with CardioClusters at day 1 (**e**), day 3 (**f**), and week 1 (**g**). **h–k** Antibody labeled immunofluorescent images from MI hearts injected with CardioClusters at week 1 (**h**), week 4 (**i**), week 12 (**j**), and week 20 (**k**). **h** Right panels show higher magnification of areas with white dotted boxes. **j** Right panel shows higher magnification of cCICs shown within white dotted box. **k** Right panels show higher magnification of cCICs and EPCs shown within white dotted boxes. Antibody labeling: anti-GFP labels cCIC, anti-mCherry labels EPC and MSC (**e**); anti-GFP labels cCIC, anti-myc labels EPC and anti-HA labels MSC (**f–h**). Scale bars: 1 mm (**a**); 100 μm (**b–h**).

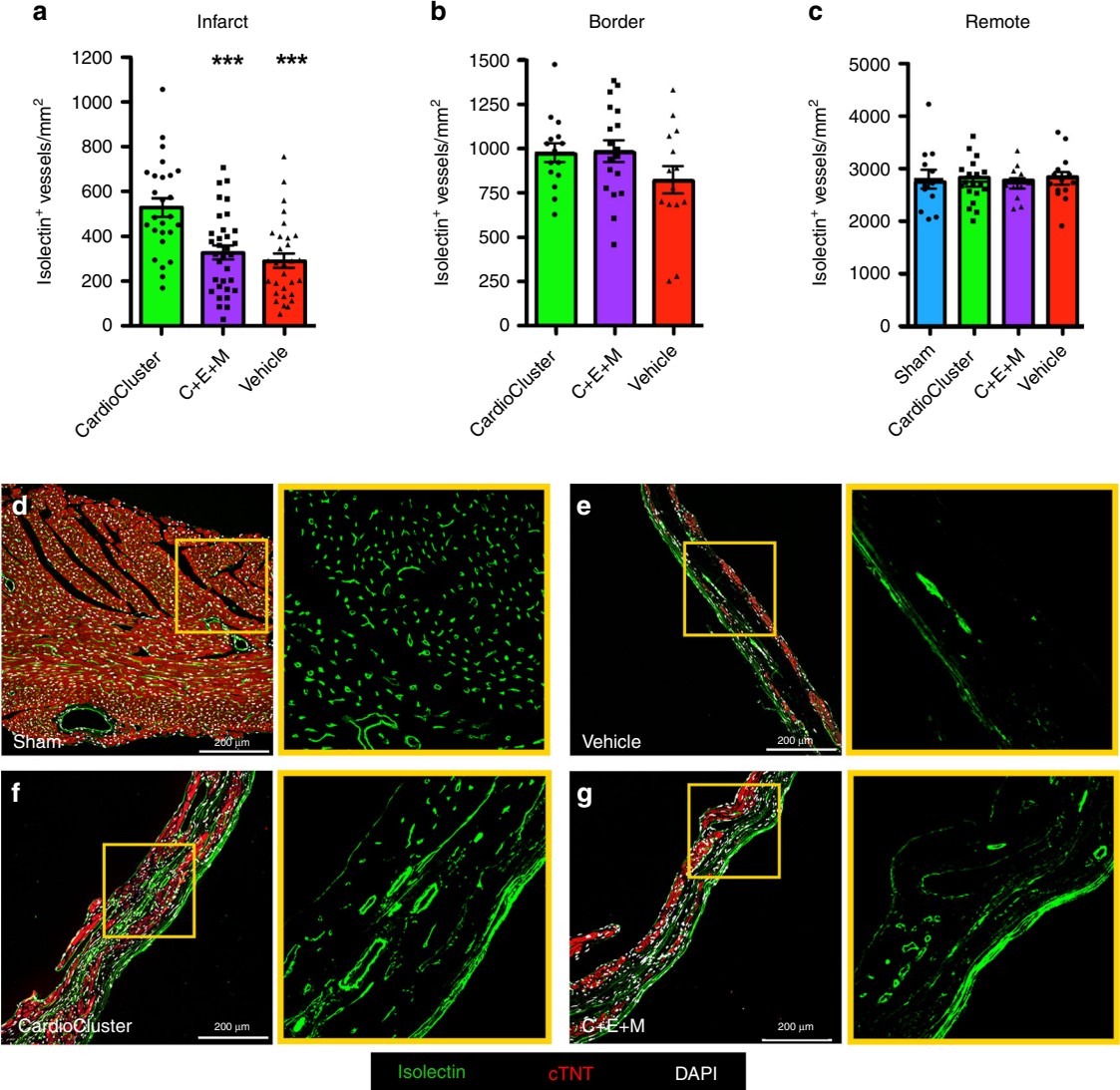

**Fig. 7 CardioCluster treatment increases capillary density in the infarct region. a–c** Quantitative analysis represents measurement of capillary density in the infarct (**a**), border zone (**b**), and remote (**c**) heart regions. Data represents mean ($n = 5$ CardioCluster-treated mice; $n = 6$ C + E + M-treated mice; $n = 5$ vehicle-treated mice; $n = 4$ sham mice) ± SEM. Data are presented as one-way ANOVA with Tukey's multiple comparisons test, ***$p < 0.001$, CardioCluster versus vehicle. ***$p < 0.001$, CardioCluster versus C + E + M. **d–g** Representative image of isolectin+ vessels in sham (**d**) and infarct regions of Vehicle (**e**), CardioCluster (**f**), and C + E + M (**g**) used to quantitate capillaries (isolectin B4; green), cardiac troponin T (cTNT; red), and nuclei (DAPI; white). Right panels show higher magnification of isolectin+ vessels from areas highlighted by yellow boxes. Scale bars, 200 μm. Source data are provided as a Source data file.

injury to blunt progression of heart failure after MI. Individual cardiomyocyte cross-sectional area was traced (Fig. 8a and Supplementary Fig. 10a–c) along with average cardiomyocyte cross-sectional area for infarct, border zone, and remote regions (Supplementary Fig. 10d–f). Treatment groups receiving either CardioClusters or the C + E + M mixed population both exhibited normalized cardiomyocyte size in the infarct region nearly identical to uninjured sham control hearts at 20 wpi (Fig. 8a). Cardiomyocytes within the border zone proximal to infarction or in remote regions from the injury site were significantly smaller in the CardioCluster treatment group compared to either C + E + M or vehicle only control groups ($p < 0.001$; Fig. 8b, c, e–g). Indeed, cardiomyocyte size in remote regions was normalized in the CardioCluster group to values similar with non-injured controls (Fig. 8c, d). Collectively these data validate the action of CardioCluster treatment to blunt hypertrophic cellular

enlargement better than dissociated mixed cell preparation or vehicle only control groups.

**CardioCluster transcriptionally resemble endogenous interstitial cells.** To understand the molecular mechanisms underlying the improvement in heart function with CardioCluster injection, we performed transcriptional profiling on monolayer cultured parental cells and CardioClusters. Given that CardioClusters are a heterogeneous cell population by design, single-cell RNA sequencing (scRNA-seq) was employed to reveal cellular transcriptome heterogeneity within the CardioCluster (Supplementary Fig. 11) at a level of resolution not achievable with bulk population analysis. Quality control testing validated parameters of cell size distribution, sequence alignment and filtering of multiplets and dying cells (Supplementary Fig. 12). Dimensionality reduction by t-SNE reveals segregation of CardioClusters

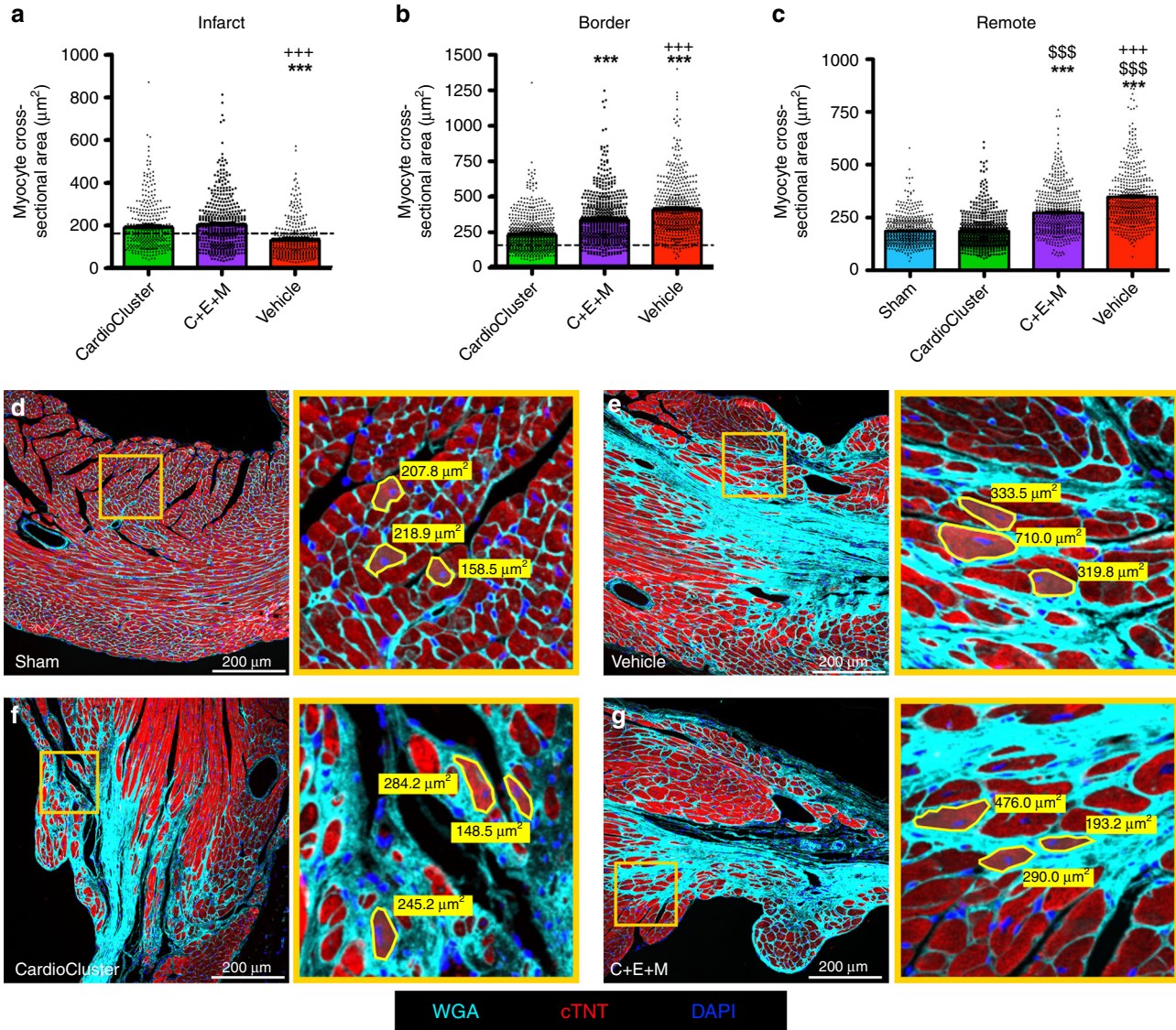

**Fig. 8 CardioCluster treatment preserves cardiomyocyte size. a–c** Quantitative analysis represents measurement of cross-sectional area of cardiomyocytes from infarct (**a**), border zone (**b**), and remote (**c**) heart regions. Dashed line: mean cross-sectional area of sham group. Data represents mean ($n = 5$ CardioCluster-treated mice; $n = 5$ C + E + M-treated mice; $n = 5$ vehicle-treated mice; $n = 4$ sham mice) ± SEM. Data are presented as one-way ANOVA with Tukey's multiple comparison test, ***$p < 0.001$, versus CardioCluster. +++$p < 0.001$, versus C + E + M. $$$$p < 0.001$, versus sham. **d–g** Representative image from sham (**d**) and border zone regions of vehicle (**e**), CardioCluster (**f**), and C + E + M (**g**) used to quantitate cardiomyocyte cross-sectional area (wheat germ agglutinin [WGA]; cyan), cardiac troponin T (cTNT; red), and nuclei (DAPI; blue). Right panels show higher magnification of areas highlighted by yellow boxes. The area of three traced cardiomyocytes per region are shown within magnified view. Scale bars, 200 μm. Source data are provided as a Source data file.

(orange cluster) distinguished by a unique transcriptome profile separating them from their constituent parental populations (red, green, blue clusters representing MSCs, cCICs and EPCs respectively; Fig. 9a). The total number of differentially expressed genes (DEGs, fold discovery rate [FDR] ≤ 0.05 and log$_2$(fold-change) ≥ ±1) increased in the CardioCluster environment (620) relative to cCIC (296), EPC (167), or MSC (211) (Fig. 9b, Supplementary Data 1). This in-silico analysis revealed that CardioClusters are a transcriptionally unique population diverged from parental cell lines cultured in 2D conditions.

Cellular identities of the three constituent parental lines comprising a CardioCluster are consistent with transcripts observed for each cell type. EPCs highly express ECSCR, ESM1, EGFL7, and RAC2, which are endothelial-related genes important for neovasculature and the angiogenic response (Fig. 9b, c). The

highly specific marker vascular endothelial statin (VE-statin) referred to as EGFL7, exhibits near-exclusive expression and action upon endothelial cells[60] and is highly expressed in EPCs (Fig. 9b). MSC-enriched transcripts include the cell surface marker THY1 (also referred to as CD90) as well as Smooth muscle α-2 actin (ACTA2) (Fig. 2b, c). Gene ontology analysis of MSCs reveals expression of extracellular matrix and adhesion molecules such as COL1A2, TIMP3 and FN1 (Fig. 9b, d). In accordance with the earlier observation that MSCs have a preference for hypoxic environments, thus migrating to the CardioCluster core (Supplementary Videos 1 and 2), MSCs are enriched for HIF1A a transcription factor that plays a key role in response to hypoxic stimuli (Fig. 9b). Lastly, transcripts associated with cell proliferation and anti-apoptotic activity such as BIRC5 and HMGB1 are differentially expressed in cCICs, as

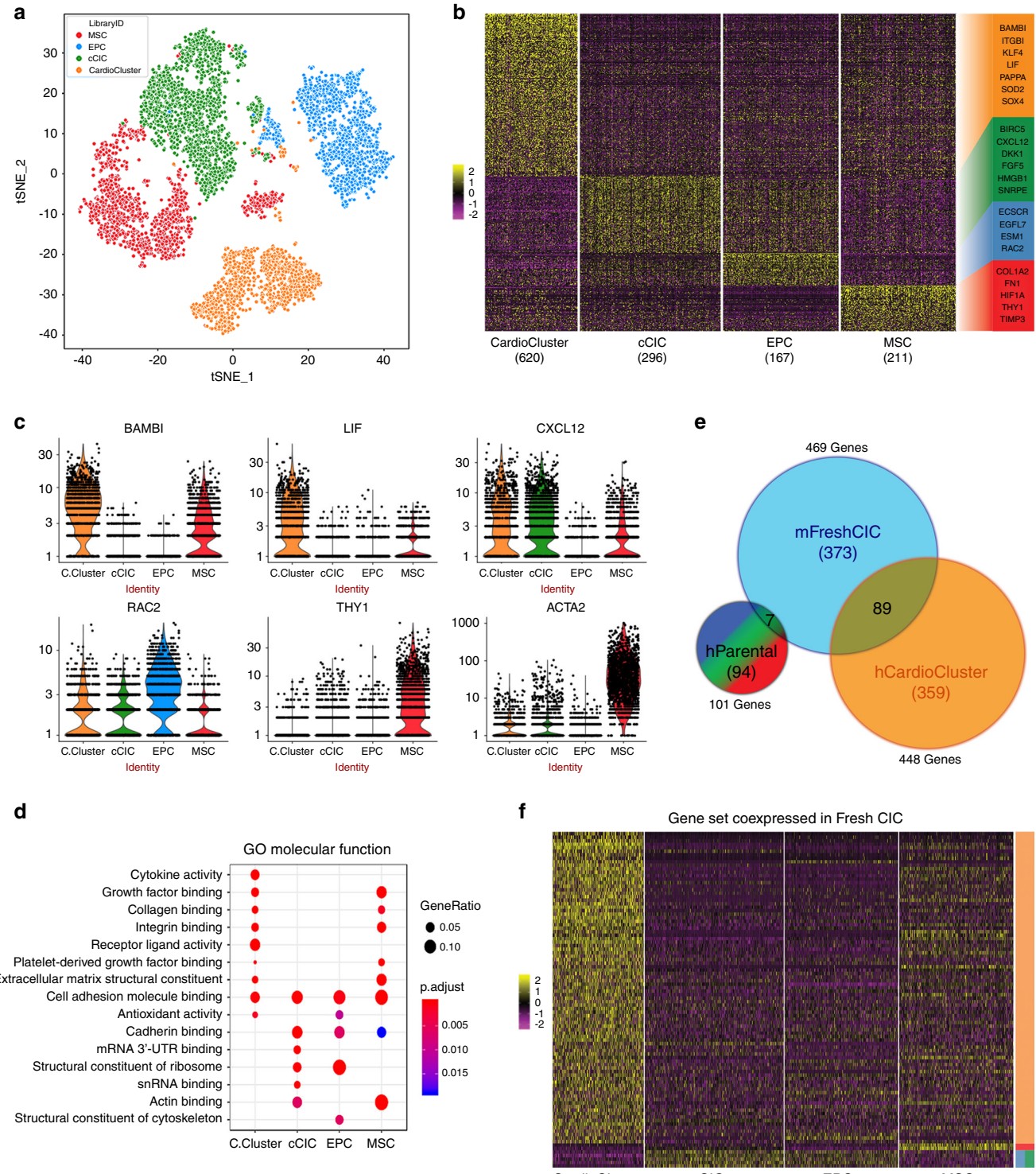

**Fig. 9 3D clustering restores cultured cells to a state resembling freshly isolated CICs revealed by single-cell RNA sequencing. a–f** Transcriptional profiling on CardioClusters and parental cells using scRNA-seq. **a** t-SNE map showing cells grown within a 3D CardioCluster (orange) predominately cluster together, while cCICs (green), EPCs (blue), and MSCs (red) primarily cluster into their own individual groups. **b** Heatmap of differentially expressed genes (DEGs) among CardioClusters and parental cells. Selected DEGs for each group are color-coded and shown on the right. **c** Violin plots of expression distribution for selected DEGs. **d** Gene Ontology (GO) term analysis of molecular functions that are enriched based on DEGs. **e**, **f** DEGs compared to genes expressed by freshly isolated mouse cardiac interstitial cells (mFreshCIC) represented in a Venn diagram (**e**) and heatmap of the gene set coexpressed by freshly isolated cells in comparison to CardioCluster and parental cell populations (**f**). hParental indicates human parental cells; hCardioCluster, human CardioCluster cells; BAMBI BMP and activin membrane bound inhibitor; LIF leukemia inhibitory factor; CXCL12 C-X-C motif chemokine 12 or stromal cell-derived factor 1 (SDF1); ACTA2 smooth muscle alpha (α)-2 actin; C. Cluster CardioCluster.

well as the chemotactic signaling molecule CXCL12, commonly referred to as stromal derived factor-1 (SDF-1), and developmental genes DKK1 and FGF5 (Fig. 9b). Collectively, these data confirm that the three parental populations are distinctly different from one another, with EPCs and MSCs expressing endothelial and stromal-associated genes.

Transcriptome profiling of CardioClusters by scRNA-seq reveals several features distinct from the three parental cell lines. CardioClusters are enriched for transcripts in multiple categories including stem cell-relevant factors (KLF4, NOTCH3, LIF, JAK1, SMAD7, BAMBI), adhesion/extracellular-matrix molecules (integrin-α2, laminin-γ1, type 1 collagen-α1, BMP1, MMP2) and cytokines (SOD2, SDF-1, FGF2, IGF), confirming earlier observations observed in co-culture experiments (Supplementary Fig. 6). One possible mechanism for the enhanced functional effects observed with CardioClusters is that 3D culture maintains an environment more similar to endogenous heart tissue and thus the cells within a CardioCluster maintain a more primitive in situ-like phenotype. To further investigate, DEGs previously identified by our group[24] between freshly isolated cardiac interstitial cells (FreshCICs) versus cultured cells were compared to the DEGs within a CardioCluster. Many of the aforementioned DEGs enriched in CardioClusters were also upregulated in FreshCICs (Fig. 9b–d, f, and Supplementary Data 2), suggesting that CardioClusters adopt a transcriptome profile with features reminiscent of cardiac interstitial cells present in the myocardium rather than cultured cells. Indeed, 89 out of the 448 DEGs present in CardioClusters are also found in FreshCICs (Fig. 9e, f, Supplementary Data 2), in stark contrast to the overlap with the 2D-cultured parental cell lines where only 7 DEGs are shared with FreshCICs (Fig. 9e, f, Supplementary Data 2). This finding supports that standard tissue culture causes expanded cells to lose their identity, unlike cells within a 3D environment. Consistent with this observation, the size of cells grown within a CardioCluster were smaller relative to 2D cultured parental counterparts ($p < 0.01$; Supplementary Fig. 12a) resembling a size more similar to freshly isolated cells, in line with cellular remodeling that occurs alongside topological changes[61]. Thus, 3D aggregation promotes a more native phenotype similar to endogenous or freshly isolated cardiac cells allowing the CardioClusters to be better adapted for adoptive transfer experimentation and facilitation of functional recovery. Additionally, the identified paracrine factors upregulated in CardioCluster co-culture experiments (Supplementary Fig. 6) were confirmed bioinformatically using scRNA-seq analysis revealing that such factors may play a critical role in the functional recovery observed with CardioCluster injection in vivo.

## Discussion

The preceding decade of cardiac cell therapy has produced substantial knowledge regarding optimization as well as limitations of current therapeutic interventions. In a field sometimes overshadowed by contentious debate[62–66], it is important to remember that there are also many points of consensus. Specifically, all parties agree that inefficient cell delivery to the site of injury, low cell retention and modest efficacy of retained cells are factors hampering advancement. Technical reinvention building upon prior success by incorporating 'next generation' approaches to surmount these established barriers represent the frontier of cell therapy research. CardioClusters introduced in this report are a novel and effective solution that integrates multiple cardiac-resident cell types into a single injectable product. The evolutionary advance offered by CardioClusters is mitigation of single cell delivery challenges through spherical self-assembly of a larger 3D structure, which provides enhanced retention to mediate

repair after delivery. Inspiration for CardioClusters is drawn from prior studies showing superiority of combinatorial cell therapy[18–21] and prolonged survival and functional engraftment of cells grown in 3D environments[67–70] or tissue-like constructs[27]. The multicellular structure and composition of CardioClusters represents a distinctly unique in vitro engineered platform to enhance the outcome of cellular therapeutics as demonstrated in preclinical testing using an established murine infarction injury model treated with a xenograft of human cardiac-derived cells.

CardioClusters were deliberately designed with multiple features anticipated to enhance efficacy. Among these properties, the essential combination of multiple cardiac cell types was enabled by our prior methodological studies to isolate and expand three distinct cardiac-derived interstitial cell types from patients with end-stage heart failure undergoing implantation of a left ventricular assist device (LVAD)[37]. End stage heart failure patients such as LVAD recipients represent likely candidates for interventional autologous therapy using cells derived from their own cardiac tissue. Combinatorial approaches harnessing beneficial attributes of multiple adult cell types are gaining acceptance as a method to enhance biological properties and efficacy based upon the tenets that: (1) no single cell population possesses all the requisite attributes for effective repair, and (2) both cardiomyogenic and non-cardiomyogenic cells contribute to myocardial repair and regeneration. Combining multiple cell types with complementary roles more efficiently mediates repair in preclinical experimental animal models of heart failure[18–21] and is currently being assessed in the CONCERT clinical trial with patients receiving mixtures of MSCs and cardiac progenitor cells[71]. Similarly, induced pluripotent stem cell-derived cardiomyocytes combined with vascular cells[72] or MSCs[73] potentiates myocardial repair, likely due to enhanced stimulation of endogenous repair mechanisms. Efficient isolation and expansion of three distinct cardiac-resident nonmyocyte populations brought together ex vivo to form CardioClusters is now technically feasible (Fig. 1)[37]. CardioCluster biological variability depending upon the source, condition, and pathologic state of donor tissue is an important and intriguing unresolved issue to be addressed in follow-up studies based upon the proof-of-principle provided in the present report.

Another enabling feature of CardioClusters is the profound influence of aggregation upon phenotypic and biological properties of the constituent cell populations. Specifically, CardioClusters foster a transcriptional profile more consistent with freshly isolated cardiac interstitial cells compared to their monolayer counterparts (Fig. 2). Even relatively short-term in vitro expansion of CICs in 2D monolayer culture results in loss of identity marker gene expression and decreased population heterogeneity by single cell RNA-Seq transcriptome profiling[24]. And although cells derived from typical 2D monolayer cultures are used to seed CardioClusters, the transcriptome signature of CardioCluster cells collectively resemble each other far more than original parental cells. The CardioCluster microenvironment promotes intercellular coordination initiated within a 3D environment, unlike traditional monolayer expansion[22–24]. Increased expression of collagen type I and III, integrins (ITGA2, ITGB1, ITGA11, ITGA1, ITGAV), fibronectin, and matrix remodeling enzymes (MMP-1, MMP-2, MMP-14, TIMP-1, TIMP-2) in CardioClusters is consistent with enhanced matrix remodeling capacity of cells maintained in a 3D microenvironment[74]. Transcript data for CardioClusters also showed elevated expression of Notch3 known to exert an important regulatory role in the contexts of development and tissue regeneration for maintenance of a progenitor pool and tissue homeostasis[75]. Elevated Notch expression and superiority of 3D aggregation culture is consistent

with results using pediatric cCICs cultured in 3D spheres of ~1500 cells, wherein 3D aggregated cCICs exert enhanced repair with increased notch signaling compared with their 2D counterparts in a right ventricular heart failure model[44]. Restoring endogenous and/or youthful characteristics to isolated cells expanded in vitro[76–79] may be one mechanism by which CardioClusters provide functional benefits to the collective population.

A third enabling feature of CardioClusters is cardioprotective action, particularly under conditions of environmental stress. In vitro co-culture assays are an established protocol to assess the potential of candidate cell types to inhibit cardiomyocyte death from pro-apoptotic challenge[49,80]. Superior ability of CardioClusters to blunt NRCM death relative to single parental cell types supports the rationale for culturing the mixed cell population together in a 3D configuration (Fig. 3). Animal studies have demonstrated that transplantation of exogenous cells exert cardioprotection through release of paracrine factors and, to a lesser extent, trans-differentiation into cardiac-resident cells[73,81–85]. This idea is supported by our transcriptomic and quantitative PCR analysis revealing the ability of interstitial cells to express a wide array of paracrine cytokines and factors that promote angiogenesis (VE-statin and RAC2) and cell survival (FGF, SDF-1, IGF, IL-6 and HGF). In vitro functional assays revealed that cardiac interstitial cells exerted prosurvival and proangiogenic effects. Furthermore, the secretome of CardioClusters also exerted protective effects on NRCMs. Importantly, the paracrine effects of CardioClusters were more powerful than those shown with the single parental populations. These findings establish the justification for subsequent in vivo testing and a potential paracrine mechanistic basis for CardioCluster action.

Superior restoration of structure and function following cardiomyopathic injury is evident from comparative testing with either CardioClusters or the dissociated cell mixture of C + E + M in xenogenic treatment of NOD[SCID] mice (Fig. 5). Empirical control of CardioCluster size to <160 µm (Fig. 1n, o) allowed for injection through a 30-gauge needle without dissociation into single cells. Improvement in FS and EF was observed starting at week 4 and maintained during the entirety of the 20-week study, with CardioCluster-treated animals showing significantly improved myocardial wall structure compared to C + E + M-treated animals concomitant with increased capillary density (Fig. 8) and preserved cardiomyocyte size (Fig. 9). CardioClusters persisted within the myocardial wall, with cells clearly visible up to 5 months post-injection (Fig. 7). The use of human cells necessitated the use of NOD[SCID] mice in our study. Although having a fully functioning immune system may lead to a different outcome, in view of what has been seen in previous studies the results should improve when the cells have been introduced into a model that has an active immune system. Future studies will be needed to address this point.

Meta-analysis examining cardiac stem cell (CSC) and MSC ability to treat MI in animal models found that treatment culminated in an absolute difference in EF ranging from 8 to 10.7% compared to control animals[86,87]. In comparison, by 20 weeks CardioCluster treatment showed a 24.2% increase in EF compared with vehicle treatment. Our data shows the significant 2-fold improvement possible with CardioClusters versus traditional single cell therapy approaches. "Off-the-shelf" potential of CardioClusters preserved in liquid nitrogen demonstrated high viability and structural integrity indistinguishable from non-frozen counterparts (Supplementary Fig. 4). Frozen/thawed CardioCluster efficacy remains to be tested in vivo, but the ability to mass-produce and preserve CardioClusters in frozen storage is attractive for clinical implementation.

The conceptual framework of CardioClusters offers almost infinite possibilities for modification and optimization for therapeutic use, as well as basic investigation of cellular interactions. For example, parameters including cell ratios, cell types, cluster size, and number of CardioClusters to inject are all worthy of further consideration. CardioCluster contain ~300 cells crucial for injectability through the inner diameter of a 30-gauge needle, in our study. However, a larger animal could tolerate a larger gauge needle and concomitantly scaled up CardioCluster size for a greater total number of cells to be injected. With the advent of ultra-low attachment microcavity plates with between 79 and 15,000+ wells per plate, the feasibility of scaling up production is possible for future clinical application and warrants further investigation. Alternatively, mini-CardioClusters of 50–100 cells with smaller diameter would allow a greater number of individual clusters to be injected. Ability to fine-tune CardioCluster size is a benefit distinct from traditional 3D cell aggregates such as cardiospheres where diameter is not controllable, necessitating dissociation into single cell suspensions of cardiosphere-derived cells for clinical use. Additional strategies to further enhance the CardioCluster concept could involve incorporating stem-cell derived cardiomyocytes[88] or even genetically modifying cells with pro-survival factors such as Pim-1[77] or chemokine receptor (CCR1) to enhance migration, survival and engraftment[89]. Likewise, culture modification using hypoxia to favor cell growth and blunt senescence-associated characteristics[90] may dramatically alter CardioCluster biological properties. With knowledge regarding combinatorial cell therapy at a rudimentary level[91,92] CardioCluster design will benefit from further investigation given the multiple possibilities for tweaking the system to enhance the outcome.

As with any novel technological approach, there are unknowns and limitations that need to be resolved for CardioCluster development. A benefit of CardioCluster design is the quick formation time of only 48 h from start to finish, however creation of CardioClusters necessitates that the multiple composite founder cell types must be ready for utilization in sequence within a short time frame. Since CardioClusters could be conceived using a plethora of possible cells, the time required for expansion of the parental cells may differ depending on the cell types chosen, particularly if using cells isolated from aged patients suffering from cardiomyopathic disease. Allogeneic implementation for CardioClusters may be possible with immunosuppressive agents or assembly using universal donor cells[93]. Prospective preparation and freeze storage of either parental cells or CardioClusters will help ease issues with timing for assembly and delivery. With respect to delivery, CardioClusters may present a safety concerns as microemboli if administered intravenously, so direct intra-myocardial injection will be used, which would be the preferred approach regardless to enhance efficacy[94].

This study presents the debut of CardioClusters as a "next generation" approach to improve upon established protocols using dissociated single cell preparations proven to be safe for administration to patients but of limited efficacy. Unlike other tissue engineering microfabrication approaches, the spontaneously formed CardioCluster 3D structure maximizes cellular interaction and allows for defined cell ratios, controlled size, and facilitates injectability without dissociation. This combination of features makes CardioClusters unique among current cell therapeutic approaches with demonstrated superiority over single cell mixed suspensions in mitigation of myocardial infarction damage. This initial step toward enhanced cell therapy provides a readily manipulatable platform that will benefit from further research development with the goal of potentiating cell-based therapeutic efficacy to mediate myocardial repair.

## Methods

**Human cardiac interstitial cell isolation**. NIH guidelines for human research were followed as approved by IRB review (Protocol #120686). Neonatal heart tissue

procured from post-mortem infants, with informed consent from family members, was provided by a commercial source (Novogenix Laboratories) and used for isolation of human cardiac cells. Fatty tissue was excised and remaining cardiac tissue was suspended in Basic Buffer (15 mL) and minced into 1 mm³ pieces. After mincing, tissue and Basic Buffer were collected in 50 mL Falcon tube. Digestive solution containing collagenase, type II 225 U/mg dry weight (Worthington, catalog #LS004174, Bio Corp, Lakewood, NJ) was dissolved in Basic Buffer (2–2.5 mg/mL) and incubated with tissue pieces for 1.5–2 h at 37 °C with continuous shaking. Digestion solution was refreshed at the one-hour time point and resulting suspensions were centrifuged at 350 g and resuspended in cCIC media (see Supplementary Table 1). Final suspension was filtered through a 100-µm filter (Corning, Inc., catalog #352360) followed by a 40-µm filter (Corning, Inc., catalog #352340) and centrifuged at 150 g for 2 min to collect CMs. The supernatant was collected and centrifuged at 350 g and resuspended in cCIC media and incubated overnight at 37 °C in 5% CO$_2$ incubator. The following day, cells in suspension were collected in 50 mL Falcon tube. Any cells attached were dissociated using a 1:1 mixture of Cellstripper (Corning, catalog #25-056-CI) and TrypLE Express (1X) (Thermo Fisher Scientific, catalog #12604-013). Resulting suspension was filtered through a 40-µm filter, centrifuged at 350 g, and resuspended in wash buffer (PBS plus 0.5% bovine serum albumin).

To isolate c-Kit⁺ cells, suspension was incubated with c-Kit–labeled beads (Miltenyi Biotec, catalog #130-091-332) and sorted according to the manufacturer's protocol. The c-Kit⁺ fraction was divided as such: half the population was suspended in cCIC media and the other half was suspended in EPC media (see Supplementary Table 1). The c-Kit⁻ population was further incubated with CD90/CD105–labeled beads and sorted according to the manufacturer's protocol (Miltenyi Biotec, catalog #130-096-253/130-051-201). Cells positive for CD90/CD105 were suspended in MSC media (see Supplementary Table 1). To isolate an EPC population, at 1 week the c-Kit⁺ population plated in EPC media was further sorted using CD133–labeled beads and sorted according to the manufacturer's protocol (Miltenyi Biotec, catalog #130-097-049). All cells were cultured at 37 °C in 5% CO$_2$ incubator in their respective growth media. cCIC and EPC were split 1:2 when they reached 60–70% confluency. MSC were split 1:2 when they reached 90% confluency. All cells used in this study were mid-passage (passages 5–10).

**CardioCluster formation.** CardioClusters are formed using 96-well, ultra-low attachment multiwell round-bottom plates (Corning, catalog #CLS7007) in a two-step process. Step 1 generates the inner core composed of cCICs and MSCs in a 1:2 ratio. The inner core of cCICs and MSCs is seeded in 100 µL/well MSC media for 24 h at 37 °C in 5% CO$_2$ incubator. Step 2 forms the outer EPC layer using a cell number equal to the number of cells used to create the central core. The EPCs are added in an additional 50 µL/well MSC media and incubated at 37 °C in 5% CO$_2$ for an additional 24 h until CardioCluster 3D structure has formed. The radius of a CardioCluster is ~150 µm, composed of a total of 400 ± 100 cells. The CardioCluster formation protocol can be found at Nature Protocol Exchange[95].

**Flow cytometry.** For live cell analysis, single cells were suspended in 100 µL wash buffer and incubated with primary antibody (see Supplementary Table 2 for dilutions) on ice for 30 min. Following, cells were washed with wash buffer and incubated with secondary antibody (1:100) for 20 min on ice. For fixed cell analysis, cells were suspended in 4% paraformaldehyde for 5 min at room temperature and then washed twice with wash buffer. For c-Kit analysis requiring permeabilization, cells were washed twice and resuspended in PBS plus 0.1% Triton X-100, 0.1 M Glycine for 3 min, then washed once. Fixed cells were suspended in 100 µL wash buffer and incubated with primary antibody on ice for 1 h. Following, cells were washed twice and incubated with secondary antibody (1:100) for 30 min on ice. For both fixed and live cells a total of 300 µl wash buffer was added post secondary incubation and the cells were analyzed by flow cytometry with a BD FACS Canto instrument (BD Biosciences). Unstained and isotype controls were used to establish baseline fluorescence levels. Data was analyzed by Flow Jo software (BD Biosciences). A minimum of 10,000 cell counts was analyzed.

**Quantitative reverse-transcriptase PCR and bioinformatics.** Total RNA was isolated using Quick-RNA MiniPrep kit (Zymo Research, catalog #R1055) according to manufacturer's protocol. RNA concentrations were determined using a Nanodrop 2000 spectrophotometer (Thermo Fisher Scientific, catalog #ND-2000) with 500 ng concentration of RNA used to generate cDNA using an iScript cDNA Synthesis kit (Bio-Rad Laboratories, Inc, catalog #170-8891). The amplified cDNA was diluted at a ratio of 1:100 in DNase- and RNase- free water. Reactions were prepared in triplicate using 6.5 µL cDNA (equivalent to 3.25 ng total RNA) per reaction using iQ SYBER Green (Bio-Rad Laboratories, Inc, catalog #170-8882) on a CFX Real-Time PCR Detection System (Bio-Rad Laboratories, Inc, catalog #1855201). Samples were normalized to 18S and data were analyzed by ΔΔCt method. Primer sequences are listed in Supplementary Table 3.

**Cell and CardioCluster morphology measurement.** Cardiac cell populations and CardioClusters were imaged using a Leica DMIL inverted tissue culture phase contrast microscope. Morphology was measured by tracing the outline of the cells or clusters using Image J software. The three measurements analyzed were area,

roundness, and length-to-width (L/W) ratios. L/W ratios were calculated by dividing feret/min feret measurements. For cell morphology, a minimum of 30 cells were measured per cell line. For CardioCluster morphology, a minimum of 15 CardioClusters were measured per timepoint.

**Cell proliferation assay.** Cell populations were plated in quadruplicate (2,000 cells/well) in a 96-well black flat bottom plate with 100 µL/well of their respective growth media. Cell proliferation rate was determined using a CyQUANT Direct Cell Proliferation Assay (Thermo Fisher Scientific, catalog #C35011) on days 0, 1, 3, and 5.

**Matrigel tube formation.** Growth factor reduced matrigel (Corning, catalog #356231) was used to coat a 96-well flat bottom plate (50 µl/well) and incubated for 30 min at 37 °C. Cell populations were plated in duplicate (5,000 cells/well) suspended in 100 µL per well of EPC basal medium (see Supplementary Table 1) and incubated at 37 °C in CO$_2$ incubator. Images of tubular networks were acquired using a Leica DMIL inverted tissue culture phase-contrast microscope 12 to 16 h after plating.

**Lentiviral constructs and cell transduction.** All cells used for CardioCluster formation were modified by expression of fluorescent-peptide fragment tags using a 3rd generation lentiviral vector. cCICs were modified with Lenti-PGK-eGFP (Addgene), MSCs were modified with PGK-Neptune-3XHA, EPCs were modified with PGK-mOrange-3Xmyc, all at multiplicity of infection (MOI) 25. Plasmid pLenti-PGK-eGFP was used as a backbone to sub-clone pLenti-PGK-mOrange-3xmyc and pLenti-PGK-Neptune-3xHA.

**Cell death assay.** Single cell populations and intact CardioClusters were plated in six-well dishes (30,000 cells per well) and incubated in starvation media (75% FBS depleted media) with 1% PSG for 24 h. The cells were then treated with 30 µM hydrogen peroxide for 4 h. Cells were detached and labeled with Annexin V (BD Biosciences; 1:175) and Propidium Iodide (PI;10 mg/ml) or Sytox Blue (Life Technologies; 1:1,000) to detect apoptosis and necrosis, respectively, by flow cytometry. Data was acquired on a BD FACSAria instrument (BD Biosciences) and analyzed with FACS Diva 3 software (BD Biosciences).

**CardioCluster preservation in liquid nitrogen and viability.** CardioClusters were collected, centrifuged at 150 $g$ for 2 min, resuspended in cold freezing medium (10% DMSO in growth medium), aliquoted into cryogenic storage vials, and frozen in an isopropanol chamber stored at −80 °C overnight. The following day vials were transferred to liquid nitrogen for a minimum of 24 h prior to thaw and cell death analysis with propidium iodide (10 mg/ml). Data was acquired on a BD FACSAria instrument (BD Biosciences) and analyzed with FACS Diva 3 software (BD Biosciences).

**Co-culture of neonatal rat cardiomyocytes with human cells.** Neonatal hearts from Sprague Dawley rats of both sex, aged 2 days postnatal, were excised and scissor minced prior to enzymatic digestion. Isolated neonatal rat cardiomyocytes (NRCMs) were plated in M199 media (Thermo Fisher Scientific, catalog #21157-029) with 15% FBS (Omega Scientific, Inc., catalog #FB-01) at a density of 200,000 cells per well of a 6-well culture dish. The following day, myocyte cultures were washed with PBS and incubated in M199 with 10% FBS for 24 h. The next morning, the cells were subjected to serum starvation (0.5% FBS in M199) for 24 h. After serum conditions, human cardiac cells were added to the plate at a ratio of 1:10 (cCICs, EPC, MSCs, all 3 single cells combined [C + E + M], and CardioClusters) and allowed to incubate with NRCMs for an additional 24 h in low serum conditions. Controls for NRCMs included leaving cells in 0.5% FBS M199, adding back 10% FBS M199 (Serum Rescue) or maintaining NRCMS in 10% FBS M199 for the duration of the experiment. NRCM size was visualized by staining cardiomyocytes with sarcomeric actinin (Sigma-Aldrich; 1:100 dilution) and nuclei with TO-PRO-3 iodide (Molecular Probes; 1:10,000 dilution). NRCM relative size was measured using forward scatter on a BD FACSAria instrument (BD Biosciences). Separation of NRCMs and human cardiac cells was accomplished with fluorescent cell sorting of negative cells (NRCMs) versus eGFP+, mOrange+ or Neptune+ cells. After sorting, cells were centrifuged and resuspended in RNAse buffer for isolation and quantitation of mRNA from NRCMs or human cells.

**Myocardial infarction and intramyocardial injection.** Animal protocols and experimental procedures were approved by the Institutional Animal Care and Use Committee at San Diego State University. Animals were housed in microisolator cages on ventilated racks with auto water. Ambient temperature was 70–72 °F, on a 12-h light dark cycle with automatic light control. Animals were randomized for treatments that were blinded to personnel carrying out the surgical procedures, injections, and physiological function analysis. Appropriate animal sample size was determined using sample size calculator (http://www.lasec.cuhk.edu.hk/sample-size-calculation.html). A total of 60 mice were used in this study. Inclusion criteria were set prior to study commencement and required ejection fraction to drop below 50% at the 1-week timepoint following infarction injury. Myocardial

infarctions were carried out on 8-week old NOD.CB17-Prkdc^scid/J female mice (The Jackson Laboratory, catalog #001303) under 2% isoflurane (Victor Medical, catalog #NDC 57319-474-06). The 3rd and 4th ribs were separated enough to get adequate exposure of the operating region, but the ribs were kept intact. The heart was squeezed out by pressing the thorax lightly and the left anterior descending artery (LAD) was ligated at the distal diagonal branch with a 7–0 suture. Infarction was confirmed by blanching of anterior left myocardium wall. Following ligation, either CardioClusters ($n = 17$), 2:3:1 ratio of cCIC+EPC+ MSC (C + E + M; $n = 15$), or a vehicle control (PBS plus 0.5% sodium alginate [NovaMatrix, catalog #4209001]; $n = 16$) were delivered intramyocardially at three separate sites in the vicinity bordering the blanched area. A total of 90,000 cells per heart were introduced into CardioCluster and C + E + M animals. The hearts were immediately placed back into the intrathoracic space followed by muscle and skin closure. Animals in sham group ($n = 12$) received a comparable surgical procedure without LAD ligation or injection. Each animal received 20 μL analgesic treatment with 0.3 mg/mL Buprenex (Victor Medical, catalog #12496-0757) at time of surgery and 12 h post-surgery.

**Pilot intramyocardial injection with FluoSpheres tracking.** Animal protocols and experimental procedures are approved by the Institutional Animal Care and Use Committee at San Diego State University. A total of 10 NOD.CB17-Prkdc^scid/J female mice (The Jackson Laboratory, catalog #001303) under 2% isoflurane (Victor Medical, catalog #NDC 57319-474-06) each received three intramyocardial injections of CardioClusters + FluoSpheres tracking beads (Thermo Fisher Scientific, catalog #F8815; 1:50,000 dilution) in LV wall. A total of 90,000 cells per heart were introduced into CardioCluster animals. The hearts were immediately placed back into the intrathoracic space followed by muscle and skin closure. All animals were euthanized 3-days post injection for histological examination of injection location.

**Echocardiography and speckle-tracking based strain analysis.** Transthoracic echocardiography was performed on lightly anesthetized mice under isoflurane (1.0–2.0%, Abbot Laboratories) using a Vevo 2100 (VisualSonics). Hearts were imaged in the 2D parasternal short-axis (SAX) view, and M-mode echocardiography of the mid-ventricle was recorded at the level of papillary muscles to calculate fractional shortening (FS). From the recorded M-mode images the following parameters were measured: left ventricular (LV) anterior wall thickness (AWT), LV posterior wall thickness (PWT), LV internal diameter (LVID), and LV volume in diastole (index: d) and systole (index: s). An LV-Trace of hearts imaged in the 2D parasternal long-axis (PLAX) view was performed in B-mode to calculate ejection fraction (EF).

Strain analysis was conducted using a speckle-tracking algorithm provided by VisualSonics (VevoStrain, VisualSonics). In brief, B-mode loops were selected from echocardiographic images based on adequate visualization of the endocardial border. A minimum of three consecutive cardiac cycles was selected for analysis based on image quality. Semi-automated tracing of the endocardial and epicardial borders were performed and then corrected as needed to achieve good quality tracking throughout each cine loop. Tracked images were then processed for strain measurements. Strain measurements were averaged over time resulting in curvilinear strain data points. Each long-axis view of the LV was divided into six standard anatomic segments for regional speckle-tracking based strain analysis throughout the cardiac cycle. Global peak strain values were averaged across all six segments. Regional peak strain values averaged the area of injury using segments 3, 5, and 6.

**Hemodynamic analyses.** Invasive hemodynamic data acquisition was performed with an ADVantage PV System (ADV500, Transonic Systems Inc.) using a 1.2F PV catheter (Transonic Systems Inc., catalog #FTH-1212B-4518). Animals were sedated using 1.2 mg/mL Ketamine (VetaKet CIII, Akorn Animal Health, Inc., catalog #59399-144-10), 0.5 mg/mL Xylazine (Anased, Akorn Animal Health, Inc., catalog #59399-110-20) dosed at 10 μL/g body weight. PV catheter was pre-calibrated in 0.9% saline for at least 30 min at room temperature before each measurement. PV catheter was inserted through right carotid artery and advanced into LV chamber to record changes in LV pressure and volume. Hemodynamic data analysis was performed offline by LabScribe v3 software (iWorx). Mice after catheterization were immediately subjected to heart retroperfusion.

**Tissue section preparation.** Mice were infused with heparin (Sigma-Aldrich, catalog #H3393) at 10 U/g body weight and anesthetized using 3% chloral hydrate solution (Sigma-Aldrich, catalog #C-8383) dosed at 10 μL/g body weight. Hearts were arrested in diastole with 0.1 M CdCl2+KCl and perfused with either 1% paraformaldehydes (cryosectioned hearts) or formalin (paraffin embedded hearts) for 5 min at 80–100 mmHg via retrograde cannulation of abdominal aorta. Ret-roperfused hearts were removed from the thoracic cavity and weighed prior to fixation. Hearts were fixed overnight in either 1% paraformaldehyde at 4 °C (cryosectioned hearts) or formalin at room temperature (paraffin embedded hearts). Hearts fixed for cryosectioning were dehydrated in 30% sucrose overnight at 4 °C followed by mounting in Neg50 frozen section medium (Thermo Fisher Scientific, catalog #6502) on dry ice. Tissues were cryosectioned at 20 μm thickness

at −20 °C. Formalin fixed hearts were processed for paraffin embedding and sectioned at 7 μm thickness at room temperature.

**Capillary density measurement.** Paraffin sections were immunolabeled with isolectin GS-IB4 conjugated to Alexa Fluor 568 (Thermo Fisher Scientific, catalog # I21412; 1:100 dilution) to visualize vasculature, in combination with cardiac troponin T conjugated to Alexa Fluor 488 (Biocompare, catalog #bs-10648R-A488; 1:200 dilution) and 4′,6-diamidino-2-phenylindole (DAPI). Scans consisted of infarct, border and remote regions for each heart analyzed. The analysis software on a Leica TCS SP8 Confocal Microscope and Image J software were used to quantitate the number of positive cells in each field of view. A minimum of two independent fields of view per cardiac region were measured. The area of cardiac tissue in each field of view was measured and used to normalize capillary numbers per mm². $N = 4$–6 hearts per group measured at week 20.

**Cardiomyocyte cross-sectional area measurement.** Paraffin sections were immunolabeled with cardiac troponin T conjugated to Alexa Fluor 488 to visualize cardiomyocytes, wheat germ agglutinin conjugated to 680 (Thermo Fisher Scientific, catalog #W32465; 1:500 dilution) to outline cellular membranes, and DAPI to visualize nuclei. Cardiomyocytes were measured in the infarct, border, and remote regions. Cross-sectioned cardiomyocytes with a centrally located nucleus were considered. Approximately 50 cardiomyocytes from 3 independent fields of view per heart region were measured using the SP8 TCS Leica drawing tool to trace cardiomyocyte cross-sectional area. $N = 4$–5 hearts per group measured at week 20.

**Infarct size quantitation.** Trichrome Stain (Masson) Kit (Sigma-Aldrich, catalog #HT15) was used to stain for collagen deposition in sham and infarcted hearts according to manufacturer's protocol. Staining was visualized using a Leica DMIL6000 microscope using XY stage tile scan and automatically stitched by Leica LAS X analysis software. Area of live versus dead myocardium was measured using the drawing tool in the SP8 TCS Leica Software using scar length over total LV length. Multiple heart sections from apex to mid-wall were averaged for scar quantification. $N = 4$–5 hearts per group measured at week 20.

**Single-cell RNA-seq preparation and data analysis.** To ensure samples were biologically matched, the three cardiac cell lines were thawed at the same time and passaged for both 3D CardioCluster formation and 2D time matched parental cultures. Size distribution was quantified for single cell suspensions of cCIC, MSC, EPC, and dissociated CardioClusters to verify cell size met droplet platform specifications (Supplementary Fig. 12a). Cells were loaded on a Chromium™ Controller (10x Genomics) and single-cell RNA-Seq libraries were prepared using Chromium™ Single Cell 3′ Library & Gel Bead Kit v2 (10x Genomics) following manufacturer's protocol. Each library was tested with Bioanalyzer (average library size: 450–490 bp). The sequencing libraries were quantified by quantitative PCR (KAPA Biosystems Library Quantification Kit for Illumina platforms P/N KK4824) and Qubit 3.0 with dsDNA HS Assay Kit (Thermo Fisher Scientific). Sequencing libraries were loaded at 2 pM on an Illumina HiSeq2500 with 2 × 75 paired-end kits using the following read length: 98 bp Read1, 8 bp i7 Index, and 26 bp Read2. Batch effects concerns were obviated in our analyses by mitigating strategies in our experimental design. Specifically, the following steps were taken: (1) all samples were processed in the same microfluidics chip, and (2) libraries were prepared in parallel and sequenced in the same lane in the HiSeq 4000. Libraries were aggregated and normalized using CellRanger. Final removal of unwanted sources of variation and batch effect corrections was performed using Seurat R Package (v2.3.4)[96].

Raw sequencing data was processed with the Cell Ranger pipeline (10X Genomics; version 2.0). Sequencing reads were aligned to the human genome hg19. Cells with fewer than 1000 genes or more than 10% of mitochondrial gene UMI count were filtered out and genes detected in fewer than three cells were filtered out using Seurat R Package (v2.3.4)[96]. The first 20 principal components were found to be significant to perform dimensionality reduction. Preparations derived from in vitro studies yielded 5659 barcoded cells for analysis, from which 1125, 1717, 1403, and 1414 corresponded to CardioClusters, cCICs, EPCs and MSCs, respectively. Approximately 2029 variable genes were selected based on their expression and dispersion. The first 20 principal components were used for the t-SNE projection and unsupervised clustering[96]. Differential expression analysis was done using Wilcoxon rank sum test and selecting for an adjusted $p$-value ≤ and a log (FC) >0.25. Global differential expression analysis was done using Loupe Cell Browser 2.0.0. Gene ontology analysis was performed using R package clusterProfiler[97].

**Statistics and reproducibility.** All statistics and posttest comparisons are stated in legends. Data are expressed as mean ± SEM. Statistical analysis was performed using GraphPad Prism (GraphPad software). A $p$-value of < 0.05 was considered statistically significant. Micrograph images presented in Fig. 1a–c are representative images from one of over 100 successful human cell isolations over a 8-year period. The CardioCluster images in Fig. 1l, m are representative of over 30 images. The brightfield images in Fig. 3c, d are representative images from three cellular treatments. The fluorescent images in Fig. 6a are representative images from 1 of 10

mice examined during the pilot murine study. The fluorescent images in Fig. 6b–h are representative images from 7 of 17 mice in the CardioCluster-treatment group examined during the myocardial infarction murine study. The images in Fig. 7d–g are representative of images from 4–6 mice per treatment group, a minimum of three images per region. The images in Fig. 8d–g are representative of images from 4–5 mice per treatment group, a minimum of three images per region. For Supplementary Fig. 1a the results are representative of three experiments. For Supplementary Fig. 2e the video still images are representative of 3 CardioCluster formation videos. The brightfield images in Supplementary Fig. 4c are representative images from three cellular treatments.

**Reporting summary**. Further information on research design is available in the Nature Research Reporting Summary linked to this article.

## Data availability

All sequencing data generated in this study have been uploaded to the Gene Expression Omnibus (GEO) database (accession number GSE133832). Datasets for freshly isolated CICs have been previously published by our group[24] and are available at the GEO database (accession number GSE114280). The data supporting this study are available from the corresponding author on reasonable requests. Source data are provided with this paper.

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

## Acknowledgements

M.M.M. is supported by NIH grant R01HL122525, Rees-Stealy Research Foundation Phillips Gausewitz, M.D., Scholars of the SDSU Heart Institute, Achievement Rewards for College Scientists (ARCS), and Inamori Fellowship. M.A.S. is supported by NIH grants: R01HL067245, R37HL091102, R01HL105759, R01HL113647, R01HL117163, P01HL085577, and R01HL122525, as well as an award from the Fondation Leducq. We would also like to gratefully acknowledge Dr. Roland Wolkowicz, Cameron Smurthwaite and Ruby Cotton for their assistance with cell sorting and flow cytometry analysis.

## Author contributions

M.M.M. and M.A.S. designed experiments. M.M.M., K.S.W., B.J.W., Z.R.Z., R.A.Jr., O.E., S.S., A.N., N.A.G., and K.F. performed experiments and analyzed data. M.M.M. and M.A.S. wrote the paper. All authors read and approved the final paper.

## Competing interests

The authors declare no competing interests.
