## [Peer Review File · Nature Communications]

Reviewers' Comments:

Reviewer #1:

Remarks to the Author:

This is an interesting study about developing a new cell product for treating heart failure. I will focus my review on the aspects related to scRNA-seq as suggested by the editor.

I am surprised by the relative homogeneity of CardioCluster (CC) cells. I expect the 3 parental cell populations can be distinguished within CC, unless somehow they all "reprogram" or "fuse" into a common cell type. If so (unlikely), how mixing different cell types achieve cellular homogeneity? And if not, the scRNA-seq analysis itself might be the problem.

scRNA-seq is prone to batch-to-batch variations. The apparent homogeneity within CCs (even though they comprise of 3 distinct cell types) and that they are well separated from the 3 parental lines show the features of batch effects. In fact, the CellRanger summaries in Suppl Fig 5 C-F hint that CC and its 3 parental lines were sequenced at different times (hence 4 of them). It is not clear to me whether batch effect correction has been performed.

The apparent homogeneity of CC cells significantly limited the usefulness of doing a scRNA-seq – a bulk RNA-seq will suffice. And it seems like the authors were treating the dataset that way.

Because MSC, EPC and cCIC have their own individual fluorescent labels, each of them can be identified within CC in the scRNA-seq, and then they can be compared with the parental line. This will help showing how the CC environment influence each cell types, which is probably one of the most important aspects of the study.

In Fig 2E, it is not clear where the mFreshCIC dataset comes from.

Reviewer #2:

Remarks to the Author:

This manuscript from the Sussman lab is well written and based on combining different cell populations in order to improve efficacy of cellular therapy (non-cardiomyocyte) following myocardial infarction. Improving outcomes following myocardial infarction is one of the most important research areas for cardiovascular disease and is therefore of interest and is important for the field.

This paper nicely demonstrates that the CardioClusters (CCs) can help cardiomyocytes survive oxidative stress in vitro and improve outcomes in vivo following myocardial infarction (function, hypertrophy, fibrosis and vascularization - all likely due to improved cardiomyocyte survival). However, there are a few critical points that my support for publication in a high impact journal such as Nature Communications:

- the study is novel but also incremental in comparison to the groups previous study (<https://doi.org/10.1161/CIRCRESAHA.115.306838>).
- it is unclear why the c-kit cells alone lead to far better functional recovery in their previous study (<https://doi.org/10.1161/CIRCRESAHA.115.306838>) but not this study. In fact, in the previous study the functional recovery (difference between control and treatment) was very similar to the CC delivery presented in this manuscript.
- the study was performed in NOD SCID mice (a necessity due to the human cells), but given the immune response to cellular therapy has recently been identified as one of the mechanisms of action when delivering non-cardiomyocyte cellular therapies it may lead to very different outcomes in organisms with a full immune system.
- the mechanism of action for the paracrine effects should be identified and confirmed with functional studies for a high impact publication such as Nature Communications to increase the

novelty. I would normally not be as critical of this, but given there vast literature already available in this field of research I feel it is more necessary than usual.

Reviewer #3:

Remarks to the Author:

In this paper, the authors described a new cell-therapy strategy to treat heart failure. They isolated and expanded the three distinct cardiac-derived interstitial cells, which were used to form scaffold-free 3D CardioClusters by a two-step process. The authors compared the transcriptional profiling of CardioClusters and their monolayer counterparts, demonstrated the unique population of CardioClusters diverged from parental cell line. CardioClusters showed elevated expression of paracrine gene and enhanced resistant capability to oxidative stress. The authors further showed the improved protective effect on cardiomyocytes in in vitro condition. Moreover, the CardioClusters improved the myocardial structure and function in a murine infarction model, evidenced by increases in capillary density, preservation of cardiomyocytes size, reduced scar size and so on. Overall, the experiments are robust, the paper is well written and the results are well described. Nevertheless, there are several questions to be addressed.

Major:

1. Since one of this work's main topics is dealing with improvement of the cell retention and survival after transplantation. The authors need to introduce and discuss more related works. For example, Moon et al. have compared the single cells and the cluster of hESC-CMs in engraftment in infarct hearts. The cluster demonstrated prolonged survival and functional engraftment with the infarcted myocardium[1]. Dr. Sawa and Dr. Shiba and their colleagues have transplant the hiPS-derived cardiac cell or tissue into infarcted primate hearts, large amount of transplanted cells were evident within the rat and monkey heart after transplantation[2,3]. In another study, cardiovascular progenitor cells were injected into the in primates with myocardial infarction and survived for around 28 days [4].

2. The reviewer has concerns over the mass production potential and the "off-the-shelf" feasibility of the CardioClusters. After myocardial infarction, around 4×10^9 cardiomyocytes could be lost [5], which means considerably amount of cells would be needed to restore the heart function. In Fig. 5-7, although the protective effect of CardioCluster is mainly from paracrine effect, somehow the small cell number of CardioCluster (~1000 cells per cluster) and the low production capability (96 cluster from one plate) would decrease the potential in future clinical application. The authors are suggested to discuss this in the updated manuscript.

3. As for the single cell transcriptional profiling, the authors found that the CardioClusters distinguishing themselves as a transcriptionally unique population diverged from parental cell lines. This single cell analysis is based on analysing the mixed single cell from CardioClusters. On the other hand, in Fig. 1L, the different types of cells within CardioCluster are actually still maintaining their original "types". The creating of the cluster is more like that different type of cells are bonded together by extra-cellular matrix such like collagen, rather than "averaging" them. So, the reviewer suggests the single cells analysis should be carried out on single cells that are dissociated from CardioClusters and sorted with FACS (with three different markers). This will give more accurate detail regarding the transcriptional profiling, which would be of interest for the board audience of Nature Communications.

4. In supplemental Figure 5a, the authors mentioned that the cells from CardioCluster are smaller than the 2D cultured parental cells. Nonetheless, it seems that the CardioCluster cells in Figure 4D have similar size like the 2D cells (Figure 4C). Please explain the possible reason for this.

Moreover, the diameter of CardioCluster did not change significantly during the 7 day's culture (Figure 1N), indicating the cell number may not increase as significantly as their 2D counterparts

(5~20 fold). It would give more insight to the cellular activity by checking the proliferation speed of cells within CardioCluster. It will be better to show the cell structure of inside CaridoCluster by SEM.

5. During the oxidative stress test (Figure 4D), It's unclear how the CardioCluster cells are plated on the 6-well dish? Have the CardioCluster undergone dissociation before plating or the cluster was directly seeded on the dish? If it's in the second condition, the surrounding cells around the cluster could be mainly derived from the EPC cells (shell of the cluster). For those surrounding cells the culture condition is actually 2D. In order to compare the cells condition in 2D and 3D culture, it could be more appropriate to just culture the CardioCluster on low attachment dish during the oxidative stress test. This could also be applied on the test for comparing the cluster after/without liquid nitrogen storage. (Supplemental Figure 4)

6. The authors have demonstrated the effective treatment by CardioCluster with a duration of 20 week. However, the trace of CardioCluster cells could only be found at day 28 after injection (Figure 7F). Have the authors checked the engraftment of CardioCluster cells in animal heart with longer time? (e.g., 10, 20 week) Moreover, it seems that the CardioCluster found in MI heart is not close to the MI area (Figure 7A) and the wall of ventricular is not as thin as it should be. How does the paracrine factor take effect if the CardioClusters are too far away from the MI area? Please discuss this.

Minor:

1. The statistic data (n and p value) is missing in Supplemental Figure 2D.

2. The statistic number (n) and scale bar for Supplemental Figure 4 are missing. The sequence of the legend is also wrong for B and C figure.

3. The H₂O₂ concentration description in the legend and the method part are different, please verify. (Figure 4)

4. Is the result in supplemental Figure 6 from single run? It's difficult to

5. The using of Dil tracking beads is missing in methods part.

6. The format of reference part is not unified.

Reference:

[1] S.-H. Moon, S.-W. Kang, S.-J. Park, D. Bae, S.J. Kim, H.-A. Lee, K.S. Kim, K.-S. Hong, J.S. Kim, J.T. Do, The use of aggregates of purified cardiomyocytes derived from human ESCs for functional engraftment after myocardial infarction, *Biomaterials* 34(16) (2013) 4013-4026.

[2] J. Li, I. Minami, M. Shiozaki, L. Yu, S. Yajima, S. Miyagawa, Y. Shiba, N. Morone, S. Fukushima, M. Yoshioka, S. Li, J. Qiao, X. Li, L. Wang, H. Kotera, N. Nakatsuji, Y. Sawa, Y. Chen, Liu, L. Human Pluripotent Stem Cell-Derived Cardiac Tissue-Like Constructs for Repairing of the Infarcted Myocardium, *Stem Cell Reports* 9(5) (2017) 1546-1559

[3] Y. Shiba, T. Gomibuchi, T. Seto, Y. Wada, H. Ichimura, Y. Tanaka, T. Ogasawara, K. Okada, N. Shiba, K. Sakamoto, Allogeneic transplantation of iPS cell-derived cardiomyocytes regenerates primate hearts, *Nature* 538(7625) (2016) 388.

[4] K. Zhu, Q. Wu, C. Ni, P. Zhang, Z. Zhong, Y. Wu, Y. Wang, Y. Xu, M. Kong, H. Cheng, Lack of remuscularization following transplantation of human embryonic stem cell-derived cardiovascular progenitor cells in infarcted nonhuman primates, *Circulation research* 122(7) (2018) 958-969.

[5] C.E. Murry, H. Reinecke, L.M. Pabon, Regeneration gaps: observations on stem cells and

cardiac repair, J. Am. Coll. Cardiol. 47(9) (2006) 1777-1785.

11/22/2019

Point-by-point response to referee comments.

Reviewer #1:

This is an interesting study about developing a new cell product for treating heart failure. I will focus my review on the aspects related to scRNA-seq as suggested by the editor.

We appreciate and highlight that this reviewer found our study “interesting” and raised reasonable points of clarification for transcriptomics results.

1. I am surprised by the relative homogeneity of CardioCluster (CC) cells. I expect the 3 parental cell populations can be distinguished within CC, unless somehow they all “reprogram” or “fuse” into a common cell type. If so (unlikely), how mixing different cell types achieve cellular homogeneity? And if not, the scRNA-seq analysis itself might be the problem.

Transcriptome profiling previously published by our group documented changes in the transcriptional heterogeneity consequential to culture adaptation to monolayer conditions¹. Transcriptional reprogramming of parental monolayer populations reflect adaptation to the three-dimensional CardioCluster environment and multifaceted cell-to-cell communication. As the reviewer observed the CardioCluster transcriptomes are distinct from parental populations, but such segregation does not lead to CardioCluster homogeneity. Analysis of single cell transcriptome of CardioClusters alone revealed four distinct sub-clusters (Fig. 1a tSNE) with respective differentially expressed genes (DEGs) (Fig. 1b Heatmap). Together, these results confirm CC internal heterogeneity and contextualizes CC transcriptome to the parental populations.

2. scRNA-seq is prone to batch-to-batch variations. The apparent homogeneity within CCs (even though they comprise of 3 distinct cell types) and that they are well separated from the 3 parental lines show the features of batch effects. In fact, the CellRanger summaries in Suppl Fig 5 C-F hint that CC and its 3 parental lines were sequenced at different times (hence 4 of them). It is not clear to me whether batch effect correction has been performed.

Batch effects concerns were obviated in our analyses by mitigating strategies in our experimental design. Specifically, the following steps were taken: 1) all samples were processed in the same microfluidics chip, and 2) libraries were prepared in parallel and sequenced in the same lane in the HiSeq 4000. Libraries were aggregated and normalized using CellRanger (Library aggregation summary shown in Fig. 2). Final removal of unwanted sources of variation and batch effect

Figure 1. CardioCluster transcriptional heterogeneity revealed by unsupervised clustering and differential expression analysis. a) t-SNE of CardioClusters internal transcriptional groups. b) Heatmap of DEGs.

corrections were performed using Seurat 2.3.4. These aspects of experimental design have been incorporated into Methods section (Page 9, Line 11-17).

3. *The apparent homogeneity of CC cells significantly limited the usefulness of doing a scRNA-seq – a bulk RNA-seq will suffice. And it seems like the authors were treating the dataset that way. Because MSC, EPC and cCIC have their own individual fluorescent labels, each of them can be identified within CC in the scRNA-seq, and then they can be compared with the parental line. This will help showing how the CC environment influence each cell types, which is probably one of the most important aspects of the study.*

As mentioned in introductory comments from this reviewer, the main objective of the submitted work was to test efficacy of CardioClusters as a therapeutic platform in the wake of cardiac injury. The depth and volume of biological information obtained from our single cell transcriptional datasets is large. Therefore, in our initial submission we aimed to present key findings within the scope of this study. We appreciate this opportunity to incorporate results not presented in the original submission and which properly reflects the usefulness of our scRNA-Seq datasets. The following two approaches were used to explore the dynamic relationship between the monolayer parental population and the 3D CardioClusters:

- 1) As pointed out by the reviewer, parental cells labeled with the fluorescent tag are expected to express the respective fluorescent transcript. We tracked the transcriptional similarities between individual cells of parental and CardioCluster origin, by incorporating and annotating the

Figure 2. Cell Ranger summary of library aggregation post-normalization.

Figure 3. Transcriptional signatures of parental populations revealed by differential expression analysis. Heatmap of DEGs.

Figure 3. Transcriptional imputation of reporter fluorophores reveals vestigial similarities between CardioClusters individual cells and parental populations.

sequences of fluorescent proteins (Orange, Neptune, GFP) into the Human Reference Genome hg19 1.2.0. Subsequent alignment on the CellRanger pipeline allowed us to detect bioinformatically the tagged cells. Computational imputation based on the expression of fluorophore confirmed vestigial transcriptional similarities in specific cells within the CardioClusters and parental MSCs (Neptune) and EPCs (Orange) (Figure 3, Imputation). Underlying similarities were not prominently observed in CICs (GFP).

- To further verify results of the imputation we performed differential expression analysis in both the parental populations (Fig. 3) and the internal populations of the CardioClusters (Fig. 1b, See response to comment 1). We hypothesized that comparing both sets of signature transcriptomes would reveal how the tridimensional environment influenced the parental populations. Cross-referencing of signature DEGs revealed vestigial intersections between the parental populations and internal CardioCluster sub-clusters (CC sC, Fig.4). Consistent with results observed in the transcriptional imputation, sub-clusters 0, 2 and 3 co-expressed genes similar to parental MSCs, while sub-cluster 1 intersected more with parental EPCs.

In conclusion, CardioCluster intersectionality with parental populations is partial, with the transcriptional sub-clusters retaining a vestigial mosaic of the primarily MSC/EPC parental populations.

4. In Fig 2E, it is not clear where the mFreshCIC dataset comes from.

We apologize for this omission. Methods have been updated with: “Datasets for freshly isolated CICs have been previously published by our group²⁵ and are available at the GEO database (accession number GSE114280).” (Page 10, line 10-12)

This data is now reflected in a newly added Supplemental Figure 5.

Citation:

[1] Kim, T. *et al.* In situ transcriptome characteristics are lost following culture adaptation of adult cardiac stem cells. *Sci Rep* 8, 12060 (2018).

Figure 4. Transcriptional intersectionality reveals vestigial similarities of CardioCluster sub-clusters (CC sC) to primarily EPC/MSC parental populations. Venn Diagram displaying overlap of transcriptional signatures.

Reviewer #2:

This manuscript from the Sussman lab is well written and based on combining different cell populations in order to improve efficacy of cellular therapy (non-cardiomyocyte) following myocardial infarction. Improving outcomes following myocardial infarction is one of the most important research areas for cardiovascular disease and is therefore of interest and is important for the field. This paper nicely demonstrates that the CardioClusters (CCs) can help cardiomyocytes survive oxidative stress in vitro and improve outcomes in vivo following myocardial infarction (function, hypertrophy, fibrosis and vascularization - all likely due to improved cardiomyocyte survival).

These collectively supportive comments are represented by recognition that the submission is “well written”, “...is of interest and is important for the field”, and “...nicely demonstrates that CardioClusters can help...” We truly appreciate such positive feedback from this reviewer.

However, there are a few critical points that my support for publication in a high impact journal such as Nature Communications:

1) The study is novel but also incremental in comparison to the groups previous study (<https://doi.org/10.1161/CIRCRESAHA.115.306838>).

The citation offered by the reviewer for this assertion is a completely unrelated concept of a CardioChimera: a single cell formed by the forced fusion of two different parental cells. The two studies are based in entirely unrelated concepts of cell fusion versus creation of a 3D CardioCluster comprised of three distinct cell types. The reviewer seems to have misunderstood the difference between these concepts.

2) It is unclear why the c-kit cells alone lead to far better functional recovery in their previous study (<https://doi.org/10.1161/CIRCRESAHA.115.306838>) but not this study. In fact, in the previous study the functional recovery (difference between control and treatment) was very similar to the CC delivery presented in this manuscript.

To clarify the results from our previous study, c-kit+ cells in the prior study failed to deliver functional recovery (EF 40.56% at week 1 fell to 37.79% at Week 18). Additionally, in our current study assessing CardioClusters we did not investigate effects of c-kit+ cells alone as an experimental group. Previous findings including the publication mentioned by the reviewer show that single cell population injections in the form of either c-Kit+ cells or MSC populations do not show substantial persistent functional improvement. It is for this reason that in the current study we performed combinatorial injection of 3 cell populations (C+E+M) as a single cell suspension versus the 3D CardioClusters.

The functional recovery shown by CC1 and CC2 (CardioChimera 1 and 2, respectively) in our previous study (<https://doi.org/10.1161/CIRCRESAHA.115.306838>) changes from 40.84% and 40.10% EF week 1 to 44.89% and 44.23% respective EF at week 18. This is approximately 4% improvement in EF. In our current study the EF goes from 27.37% week 1 to 39.73% at week 20. This is a 12.36% improvement in heart function which is over a 3-fold increase from the previous CardioChimera study.

When comparing our study to the field in general, meta-analysis provided by Zwetsloot *et al.* compiled a total of 80 papers using cell therapy to treat MI in both large and small animals¹. The pooled analysis found that cardiac stem cell treatment culminated in an absolute difference in EF of 10.7% compared with control animals. Pooled observations made from large animal studies using MSC and bone marrow mononuclear cell therapy show similar results (8.0% and 7.6%, respectively)². In comparison, at 20 weeks CardioClusters treatment showed a 24.2% difference in EF versus vehicle treatment. Our data shows the significant improvement possible with CardioClusters versus traditional single cell approaches.

In addition, numerous differences in this current study versus our previous CardioChimera study could readily account for differences in outcomes and render such comparisons of little value. Specifically, human cells are used for the CardioCluster study versus mouse cells in the previous CardioChimera study. Furthermore, as mentioned by the reviewer, mouse strains differ between the two studies. Therefore, comparisons between the two studies are not appropriate since various mouse strains respond differently to infarction challenge^{3,4}.

Citations:

1. Zwetsloot, P.P. et al. Cardiac Stem Cell Treatment in Myocardial Infarction: A Systematic Review and Meta-Analysis of Preclinical Studies. *Circ Res* **118**, 1223-1232 (2016).
2. Jansen Of Lorkeers, S.J. et al. Similar effect of autologous and allogeneic cell therapy for ischemic heart disease: systematic review and meta-analysis of large animal studies. *Circ Res* **116**, 80-86 (2015).
3. Barnabei, M.S. et al. Influence of genetic background on ex vivo and in vivo cardiac function in several commonly used inbred mouse strains. *Physiological Genomics* **42A**, 103-1113 (2010).
4. Patterson, M. et al. Frequency of mononuclear diploid cardiomyocytes underlies natural variation in heart regeneration. *Nat Genet* **49**, 1346-1353 (2017).

3) The study was performed in NOD SCID mice (a necessity due to the human cells), but given the immune response to cellular therapy has recently been identified as one of the mechanisms of action when delivering non-cardiomyocyte cellular therapies it may lead to very different outcomes in organisms with a full immune system.

As the reviewer correctly states, human cell transfer necessitated use of NOD SCID mice in our study. Although having a fully functioning immune system may lead to a different outcome, in view of what has been seen in previous studies the results should improve when cells would be introduced into a syngeneic model possessing a working immune system. Future studies will be needed to address this point. (This information has now been added to the Discussion section of the manuscript: Page 30, line 11-15)

4) The mechanism of action for the paracrine effects should be identified and confirmed with functional studies for a high impact publication such as *Nature Communications* to increase the novelty. I would normally not be as critical of this, but given there vast literature already available in this field of research I feel it is more necessary than usual.

As the reviewer points out, paracrine effects have been extensively characterized in prior studies and appear to be one aspect of a complex multifaceted response responsible for improvements reported *in vivo*. For this reason, we decided to forego a detailed redundant-type analysis that would add relatively incremental understanding to the field beyond what is already known. To address the reviewers concerns we have now cited several papers that have identified factors previously, which correspond with the factors we identified in our *in vitro* assays. We thank the review for this suggestion.

Additional studies now cited in the manuscript (page 29 and 30, line 19-23 and line 1-8) and discussed as follows:

“Animal studies have demonstrated that transplantation of exogenous cells exert cardioprotection through release of paracrine factors and, to a lesser extent, trans-differentiation into cardiac resident cells^{79, 87-93}. This idea is supported by our transcriptomic and quantitative PCR analysis revealing the ability of cardiac interstitial cells to express a wide array of paracrine cytokines and factors that promote angiogenesis (such as VE-statin and RAC2) and cell survival (such as FGF, SDF-1, IGF, IL-6 and HGF). *In vitro* functional assays revealed that cardiac interstitial cells exerted prosurvival and proangiogenic effects. Furthermore, the secretome of CardioClusters also exerted protective effects on neonatal rat cardiomyocytes. Importantly, the paracrine effects of CardioClusters were more powerful than those shown with the single parental populations. These findings establish the justification for subsequent *in vivo* testing and a potential paracrine mechanistic basis for CardioCluster action.”

1. Haider, H., Jiang, S., Idris, N.M. & Ashraf, M. IGF-1-overexpressing mesenchymal stem cells accelerate bone marrow stem cell mobilization via paracrine activation of SDF-1alpha/CXCR4 signaling to promote myocardial repair. *Circ Res* **103**, 1300-1308 (2008).
 2. Jackson, R. *et al.* Paracrine Engineering of Human Cardiac Stem Cells With Insulin-Like Growth Factor 1 Enhances Myocardial Repair. *J Am Heart Assoc* **4**, e002104 (2015).
 3. Mayfield, A.E. *et al.* Interleukin-6 Mediates Post-Infarct Repair by Cardiac Explant-Derived Stem Cells. *Theranostics* **7**, 4850-4861 (2017).
 4. Forte, G. *et al.* Human cardiac progenitor cell grafts as unrestricted source of supernumerary cardiac cells in healthy murine hearts. *Stem Cells* **29**, 2051-2061 (2011).
 5. Oskouei, B.N. *et al.* Increased potency of cardiac stem cells compared with bone marrow mesenchymal stem cells in cardiac repair. *Stem Cells Transl Med* **1**, 116-124 (2012).
-

Reviewer #3:

In this paper, the authors described a new cell-therapy strategy to treat heart failure. They isolated and expanded the three distinct cardiac-derived interstitial cells, which were used to form scaffold-free 3D CardioClusters by a two-step process. The authors compared the transcriptional profiling of CardioClusters and their monolayer counterparts, demonstrated the unique population of CardioClusters diverged from parental cell line. CardioClusters showed elevated expression of paracrine gene and enhanced resistant capability to oxidative stress. The authors further showed the improved protective effect on cardiomyocytes in in vitro condition. Moreover, the CardioClusters improved the myocardial structure and function in a murine infarction model, evidenced by increases in capillary density, preservation of cardiomyocytes size, reduced scar size and so on. Overall, the experiments are robust, the paper is well written and the results are well described. Nevertheless, there are several questions to be addressed.

Major:

1. Since one of this work's main topics is dealing with improvement of the cell retention and survival after transplantation. The authors need to introduce and discuss more related works. For example, Moon et al. have compared the single cells and the cluster of hESC-CMs in engraftment in infarct hearts. The cluster demonstrated prolonged survival and functional engraftment with the infarcted myocardium [1]. Dr. Sawa and Dr. Shiba and their colleagues have transplant the hiPS-derived cardiac cell or tissue into infarcted primate hearts, large amount of transplanted cells were evident within the rat and monkey heart after transplantation [2,3]. In another study, cardiovascular progenitor cells were injected into the in primates with myocardial infarction and survived for around 28 days [4].

We thank the reviewer for this suggestion, additional citations were introduced and discussed in the manuscript as detailed below together with the corresponding reference as provided by the reviewer:

Reference:

[1] S.-H. Moon, S.-W. Kang, S.-J. Park, D. Bae, S.J. Kim, H.-A. Lee, K.S. Kim, K.-S. Hong, J.S. Kim, J.T. Do. The use of aggregates of purified cardiomyocytes derived from human ESCs for functional engraftment after myocardial infarction, *Biomaterials* 34(16) (2013) 4013-4026.

Appears as reference **27** on page 27, lines 5-6: "Inspiration for CardioClusters is drawn from prior studies showing superiority of combinatorial cell therapy¹⁹⁻²² and prolonged survival and functional engraftment of cells grown in 3D environments⁷³⁻⁷⁶ or tissue-like constructs²⁷."

[2] J. Li, I. Minami, M. Shiozaki, L. Yu, S. Yajima, S. Miyagawa, Y. Shiba, N. Morone, S. Fukushima, M. Yoshioka, S. Li, J. Qiao, X. Li, L. Wang, H. Kotera, N. Nakatsuji, Y. Sawa, Y. Chen, Liu, L. Human Pluripotent Stem Cell-Derived Cardiac Tissue-Like Constructs for Repairing of the Infarcted Myocardium, *Stem Cell Reports* 9(5) (2017) 1546-1559.

Appears as reference **76** on page 27, lines 5-6: "Inspiration for CardioClusters is drawn from prior studies showing superiority of combinatorial cell therapy¹⁹⁻²² and prolonged survival and functional engraftment of cells grown in 3D environments⁷³⁻⁷⁶ or tissue-like constructs²⁷."

[3] Y. Shiba, T. Gomibuchi, T. Seto, Y. Wada, H. Ichimura, Y. Tanaka, T. Ogasawara, K. Okada, N. Shiba, K. Sakamoto, Allogeneic transplantation of iPS cell-derived cardiomyocytes regenerates primate hearts, *Nature* 538(7625) (2016) 388.

We would be happy to include this reference in the manuscript but were unable to identify a relevant location in the text. We welcome feedback from the reviewer as to appropriate placement for this citation.

[4] K. Zhu, Q. Wu, C. Ni, P. Zhang, Z. Zhong, Y. Wu, Y. Wang, Y. Xu, M. Kong, H. Cheng, Lack of remuscularization following transplantation of human embryonic stem cell-derived cardiovascular progenitor cells in infarcted nonhuman primates, *Circulation research* 122(7) (2018) 958-969.

Appears as reference **12** on page 4, lines 4-6: "While promising, cellular therapy has been hindered by marginal improvement in cardiac function, in part due to low engraftment and persistence of the transplanted cells¹⁰⁻¹²."

*2. The reviewer has concerns over the mass production potential and the "off-the-shelf" feasibility of the CardioClusters. After myocardial infarction, around 4*10⁹ cardiomyocytes could be lost [5], which means considerably amount of cells would be needed to restore the heart function. In Fig. 5-7, although the protective effect of CardioCluster is mainly from paracrine effect, somehow the small cell number of CardioCluster (~1000 cells per cluster) and the low production*

capability (96 cluster from one plate) would decrease the potential in future clinical application. The authors are suggested to discuss this in the updated manuscript.

CardioClusters used in this study are 400 ± 100 cells seeded as a single Cluster/well in a 96 well plate. The reasoning for this low cell number was due to the internal diameter of a 30-gauge needle. In future studies where a larger animal model is used, the size of both the needle and the CardioClusters can be scaled up. Additionally, new options for non-adherent round bottom 6-well dishes have become available (Corning cat. No. 4440) which can generate between 79 – 15,000+ CardioClusters per well. These options may ease future clinical application and warrant further investigation. This information has now been added to the Discussion: “With the advent of ultra-low attachment microcavity plates with between 79 – 15,000+ wells per plate, the feasibility of scaling up production is possible for future clinical application and warrants further investigation.” (Page 31, lines 20-22)

3. As for the single cell transcriptional profiling, the authors found that the CardioClusters distinguishing themselves as a transcriptionally unique population diverged from parental cell lines. This single cell analysis is based on analysing the mixed single cell from CardioClusters. On the other hand, in Fig. 1L, the different types of cells within CardioCluster are actually still maintaining their original “types”. The creating of the cluster is more like that different type of cells are bonded together by extra-cellular matrix such like collagen, rather than “averaging” them. So, the reviewer suggests the single cells analysis should be carried out on single cells that are dissociated from CardioClusters and sorted with FACS (with three different markers). This will give more accurate detail regarding the transcriptional profiling, which would be of interest for the board audience of Nature Communications.

We agree with the reviewer and performed single cell analysis, not bulk, specifically to circumvent this issue of “mixed” or averaged RNA content from different cell types. Instead, each cell of the CardioCluster was analyzed as an individual entity. We appreciate the reviewer’s interest in increasing the impact of our study and have now included in the revised manuscript a bioinformatics approach to expand and track the transcriptional profile of our fluorescently labeled cells. Instead of sorting the CardioCluster labeled cells using the fluorescent signal as a reporter (FACS approach), we sorted them based on the expression of fluorescent transcript followed by a computational imputation. This approach required annotation of the Human Reference Genome hg19 1.2.0 with the sequences of fluorescent proteins (Orange, Neptune, GFP) prior alignment. We have expanded and incorporated new results on the transcriptional relationship between parental populations and CardioCluster sub-clusters based on this and previous comments (See Response to Reviewer 1, comments 1 and 3). We hope you find this new information satisfactory and more appealing to the audience of *Nature Communications*.

4. In supplemental Figure 5a, the authors mentioned that the cells from CardioCluster are smaller than the 2D cultured parental cells. Nonetheless, it seems that the CardioCluster cells in Figure 4D have similar size like the 2D cells (Figure 4C). Please explain the possible reason for this.

We thank the reviewer for this question and to clarify, the cells in Figure 4 have undergone serum starvation and hydrogen peroxide treatment and thus changes in cell size may have occurred. The cells that remain attached to the tissue culture plate in Figure 4 were undergoing apoptosis and partial dissociation from the culture plate. The cell size comparison in Supplemental Figure 5A was performed on cells in standard cell culture conditions, on healthy cells that were enzymatically detached from the culture dishes, not adherent to a tissue culture plate as in the cell death experiment in Figure 4.

Supplemental Figure 5A analyzes cell size differences between 3D and 2D culture conditions. Once these 3D cultured cells are allowed to adhere on standard cell culture plates cell morphology and topography will change (Topological Arrangement of Cardiac Fibroblasts Regulates Cellular Plasticity, Jingyi Yu paper [reference 56]) this information has now been added (Page 18, lines 12-13): “Consistent with this observation, the size of cells grown within a CardioCluster were smaller relative to 2D cultured parental counterparts ($p < 0.01$; Supplemental Figure 5A) resembling a size more similar to freshly isolated cells, consistent with cellular remodeling that occurs alongside topological changes⁵⁶.” Consequently, if the cells appear similar in size in Figure 4, I would not be surprised by this because both these cell populations are on adherent dishes.

To help clarify changes were made to the Methods section: “Size distribution was quantified for single cell suspensions of cCIC, MSC, EPC and dissociated CardioClusters to verify cell size met droplet platform specifications (Supplemental Figure 6A).” (Page 9, lines 1-3)

5. Moreover, the diameter of CardioCluster did not change significantly during the 7 day's culture (Figure 1N), indicating the cell number may not increase as significantly as their 2D counterparts (5~20 fold). It would give more insight to the cellular activity by checking the proliferation speed of cells within CardioCluster. It will be better to show the cell structure of inside CardioCluster by SEM.

Supplemental Figure 2D quantitation of CardioCluster “Diameter” and “Area” on day 1, 3, and 7 shows a slight increase over this one-week time period, suggesting very limited cellular proliferation or cellular remodeling relative to 2D culture as noted by the reviewer. Indeed, low level cell cycle activity was confirmed in studies using the FUCCI cell cycle indicator that revealed few cycling cells in 3D culture (Figure 5).

Figure 5. Representative Image of FUCCI labeled cells cultured as a CardioCluster. Schematic of FUCCI labeling different phases of the cell cycle (Figure 5A). White arrows point to two cells out of the entire CardioCluster that are in the S/G2/M phase of the cell cycle (Azami Green+ [Az]; Figure 5C). The rest of the cells within the CardioClusters are in the G1 phase of the cell cycle (monomeric Kusabira-Orange+ [mKO]; Figure 5D). BF indicates Brightfield.

6. During the oxidative stress test (Figure 4D), It's unclear how the CardioCluster cells are plated on the 6-well dish? Have the CardioCluster undergone dissociation before plating or the cluster was directly seeded on the dish? If it's in the second condition, the surrounding cells around the cluster could be mainly derived from the EPC cells (shell of the cluster). For those surrounding cells the culture condition is actually 2D. In order to compare the cells condition in 2D and 3D culture, it could be more appropriate to just culture the CardioCluster on low attachment dish during the oxidative stress test. This could also be applied on the test for comparing the cluster after/without liquid nitrogen storage. (Supplemental Figure 4)

We thank the reviewer for this point. We have clarified in the Methods section that the CardioClusters were directly seeded onto the 6-well dish as follows: “**Cell Death Assay.** Single cell populations and intact CardioClusters were plated in 6-well dishes (30,000 cells per well) and incubated in starvation media (75% FBS depleted media) with 1% PSG for 24 hours. The cells were then treated with 30 μ M hydrogen peroxide for 4 hours. Cells were detached and labeled with Annexin V (BD Biosciences; 1:175) and Propidium Iodide (PI;10mg/ml) or Sytox Blue (Life Technologies; 1:1,000) to detect apoptosis and necrosis, respectively, by flow cytometry. Data was acquired on a BD FACSAria instrument (BD Biosciences) and analyzed with FACS Diva 3 software (BD Biosciences).” (Page 5, lines 10-13)

The purpose of this experiment was to mimic environmental stresses potentially encountered by CardioClusters injected following ischemic injury *in vivo*. While the surrounding shell layer is composed of both cCIC and EPC, not just EPC, this is the same layer of cells that will be exposed to the oxidative stress seen *in vivo*. The reason we did not treat the CardioClusters while in low attachment dishes was because we wanted to keep the assay parameters the same among all treatment groups, as well as mimic an *in vivo* environment where the cells will be adherent within the extracellular matrix environment of the heart.

7. The authors have demonstrated the effective treatment by CardioCluster with a duration of 20 week. However, the trace of CardioCluster cells could only be found at day 28 after injection (Figure 7F). Have the authors checked the engraftment of CardioCluster cells in animal heart with longer time? (e.g., 10, 20 week) Moreover, it seems that the CardioCluster found in MI heart is not close to the MI area (Figure 7A) and the wall of ventricular is not as thin as it should be. How does the paracrine factor take effect if the CardioClusters are too far away from the MI area? Please discuss this.

At the time of initial submission, the 28 day post-injection time point was the furthest longitudinal time assessment performed on tissue sections to demonstrate enhanced persistence. As requested by the reviewer, we have now examined hearts from 12 week and 20 week time points from hearts receiving CardioCluster delivery. Clear evidence of CardioCluster-derived cells were observed in 2 of the 3 hearts at the 12 week time point and 4 of the 6 hearts at the 20 week time point. Given that these observations were made using only a single section from each heart under examination, it is certainly possible that more CardioCluster-derived cells are present in the recipient hearts remaining to be identified. However, these robust reproducible observations reinforce the novelty and significance of CardioCluster methodology to foster enhanced engraftment and persistence relative to prior studies using dissociated cell suspensions. We have now included representative images from both time points as additional panels added to Figure 7 (as shown below in Panels G, G', H, and H'). Additional confocal microscopy examples of CardioCluster-derived cells at the 12 and 20 week time points were obtained but not included due to space limitations.

The figure legend has been modified by including these results as follows (changes in *italics*):

“Figure 7. CardioCluster show enhanced engraftment and persistence in the myocardial wall

A, Immunofluorescent tile scan of a cryosectioned heart *from an uninjured animal* injected with CardioClusters day 3 post-injection tracked by FluoSpheres with no antibody labeling required for visualization of cells. **A'-A'''**, Higher magnification of areas with white dotted boxes. **A'**, Illustrates the removal of the 555nm channel in order to better visualize cCIC (green) and MSC (blue), without FluoSpheres. **A''-A'''**, 555nm channel restored (**A''**) and field of view magnified (**A'''**). **B-D**, Immunofluorescent images from cryosectioned MI hearts injected with CardioClusters at day 1 (**B**), day 3 (**C**), and week 1 (**D**). **E-H**, Antibody labeled immunofluorescent images from MI hearts injected with CardioClusters at week 1 (**E**), week 4 (**F**), week 12 (**G**), and week 20 (**H**). **E'**, Higher magnification of areas with white dotted boxes. **G'**, Higher magnification of cCICs shown within white dotted box. **H'**, Higher magnification of cCICs and EPCs shown within white dotted boxes. Antibody labeling: anti-GFP labels cCIC, anti-mCherry labels EPC and MSC (E); anti-GFP labels cCIC, anti-myc labels EPC and anti-HA labels MSC (F-H). Scale bars: 1mm (A); 100µm (B-H).”

Additional panels G, G', H and H' added to Figure 7:

Figure 7A is from a pilot study to demonstrate successful injection and tracking of CardioClusters *in vivo*. This heart, correctly noted by the reviewer, is not infarcted. We thank the reviewer for noting this distinction and have now amended the Results section (Page 24, lines 13-15): “CardioCluster localization was tracked with co-injection of FluoSpheres tracking beads in pilot studies to confirm delivery location in tissue sections (Figure 7A).” Additionally, in the Supplemental Methods section (Page 7, lines 4-13) we have detailed our FluoSpheres tracking procedure: **“Pilot Intramyocardial Injection with FluoSpheres Tracking.** Animal protocols and experimental procedures are approved by the Institutional Animal Care and Use Committee at

San Diego State University. A total of 10 NOD.CB17-Prkdc^{scid}/J female mice (The Jackson Laboratory, catalog #001303) under 2% isoflurane (Victor Medical, catalog #NDC 57319-474-06) each received 3 intramyocardial injections of CardioClusters + FluoSpheres tracking beads (Thermo Fisher Scientific, catalog #F8815; 1:50,000 dilution) in LV wall. A total of 90,000 cells per heart were introduced into CardioCluster animals. The hearts were immediately placed back into the intrathoracic space followed by muscle and skin closure. All animals were euthanized 3-days post injection for histological examination of injection location.”

Figure 7 Legend was also modified, as shown above, to include the details that FluoSphere injections were performed on uninjured animals.

Minor:

1. The statistic data (n and p value) is missing in Supplemental Figure 2D.

Methods section and Supplemental Figure Legend 2D have been updated with the missing n and p values for CardioCluster morphological analysis as follows: “D, CardioCluster morphometric parameters measuring area (a.u.: arbitrary units.), roundness, and length-to-width (L/W) ratio over a 7 day time course for CardioClusters ranging from 100-1000 cells (n=15-20 CardioClusters measured per timepoint). Data are presented as 1-way ANOVA, *p<0.05, **P<0.01, ***p<0.001, versus day 1. #p<0.05, versus day 3.”

Updated Methods for Supplemental Figure 2: “For CardioCluster morphology, a minimum of 15 CardioClusters were measured per timepoint.”

Updated Supplemental Figure 2D:

2. The statistic number (n) and scale bar for Supplemental Figure 4 are missing. The sequence of the legend is also wrong for B and C figure.

Supplemental Figure 4 legend has been updated with the missing number (n). Scale bars have been added. The sequence order of the legend has been corrected as follows: “Supplemental Figure 4. CardioClusters frozen in liquid nitrogen maintain structural integrity and viability. **A**, Representative flow cytometry plots showing propidium iodide (PI) gating strategy used in freezing assay. **B**, Quantification of percent necrotic (PI⁺) cells from non-frozen versus liquid nitrogen frozen experimental groups (n=3 independent experiments, an entire 96-well plate of CardioClusters per experimental group). Data are presented as 1-way ANOVA, *p<0.05, **P<0.01, ***p<0.001, versus non-frozen CardioClusters. ns indicates not statically significant. **C**, Brightfield images showing cell outgrowth of non-frozen versus liquid nitrogen frozen CardioClusters. Scale bar, 100 µm.”

(Updated panel of Supplemental Figure 4C)

3. *The H2O2 concentration description in the legend and the method part are different, please verify. (Figure 4)*

We thank the reviewer for pointing out this omission. The concentration has been corrected in the Methods section to 30 μ M hydrogen peroxide.

4. *Is the result in supplemental Figure 6 from single run? It's difficult to*

Yes, the representative images from Supplemental Figure 6 were all run on the same day.

5. *The using of Dil tracking beads is missing in methods part.*

We thank the reviewer for pointing out that this experimental procedure was missing from the methods section, this information has now been added: "**Pilot Intramyocardial Injection with FluoSpheres Tracking.** Animal protocols and experimental procedures are approved by the Institutional Animal Care and Use Committee at San Diego State University. A total of 10 NOD.CB17-Prkdc^{scid}/J female mice (The Jackson Laboratory, catalog #001303) under 2% isoflurane (Victor Medical, catalog #NDC 57319-474-06) each received 3 intramyocardial injections of CardioClusters + FluoSpheres tracking beads (Thermo Fisher Scientific, catalog #F8815; 1:50,000 dilution) in LV wall. A total of 90,000 cells per heart were introduced into CardioCluster animals. The hearts were immediately placed back into the intrathoracic space followed by muscle and skin closure. All animals were euthanized 3-days post injection for histological examination of injection location." (Page 7, lines 4-13)

6. *The format of reference part is not unified.*

The format has now been corrected for *Nature* journals.

Reviewers' Comments:

Reviewer #1:

Remarks to the Author:

I am satisfied with the authors' responses as they have addressed all my concerns.

Reviewer #2:

Remarks to the Author:

While some of the critiques have been addressed, I still have reservations about publication of this study given my major critiques have not been adequately addressed.

Conceptual

Responses to Reviewer 2 pts 1, 2,3 and 4:

The caveats raised by all the rebuttal arguments are not sufficient to support the lack of identification of the mechanism. As the authors state correctly bone marrow cell, MSC and resident cardiac cell populations have led to EF improvements of around 7-11%. However, the problem is:

1. These responses can be variable lab-lab and without knowing the underlying mechanism its very difficult to troubleshoot why this is the case
2. Translation into the clinic will also require reproducibility, studies in a variety of animal models and stringent quality control of end stage production. As the mechanism is undefined this is a huge barrier to translation and would create a range of difficulties along the way.
3. In this manuscript the improvement in EF is roughly 2 fold higher than in the other cellular studies. It is paramount to discover why this is the case because this could potentially enhance the therapeutic efficacy further.
4. The authors claim that the CardioChimera study is totally distinct. I disagree with this statement and regard both studies as improving the cellular therapy by combining cell types. If the paper had additional insight into the mechanism then I could be persuaded. I do note the studies showing improved survival after oxidative stress in vitro, but there is some issues with this data too (see below). In my first review I gave the authors a chance to delve into the mechanism further and was hoping for these issues to be clarified.
5. I am not suggesting that the study does not warrant publication, but I do not think is it of the level I generally read at Nature Communications.

Major

6. Responses to Reviewer 2 pt 2:

Quoting change in EF from week 1 to week 20 is flawed if the cells are given at time of MI (which the methods suggest). If part of the mechanism is truly a reduction in CM death in the face of oxidative stress, then one would expect a big difference after 1 week as nearly all the CM death following MI will occur within this period. Apoptosis after delivery in vivo would need to be quantified.

7. Responses to Reviewer 2 pt 2:

In the previous study on CardioChimeras there does appear to be a bit of a difference at week 1 which would probably become significant with more animal numbers, followed by either declining function in controls or sustained function in some of the treatments. Given in both the CardioChimera and this CardioCluster study the cells are delivered at the time of MI (as far as the methods state) quoting the delta EF from week 1 to 18 or 20 is flawed. I was quoting the overall magnitude of the effect relative to the controls. If the authors want to study delta EF properly the experimental design needs to be altered to deliver the cells at week 1 (but I understand this would place a big stress on the animals and is the reason it was not designed this way).

8. Responses to Reviewer 2 pt 4:

The major increased factor with the CardioCluster seems to be IGF-1 (Supp Fig 8). This gets back to my mechanistic question, how does the efficacy of CardioClusters compare with delivery of say IGF-1 in this example following MI (delivered via eg. mini-pumps or over-expression using AAV

mediated delivery to the myocardium)? Approaches such as this would be a much more straightforward to translate.

Minor (potentially)

9. Review 1 pt2:

This raises a concern for me. The cells should also be biologically matched as well. The sequencing should have been run from the same initial 3 cultures (ie thawed at the same time), passaged to make both CC and 2D time matched cultures and then RNA harvested. Can the authors confirm/please add to the methods, their experimental design?

Reviewer #3:

Remarks to the Author:

I'd like to thank the authors for taking the time to address the raised questions about this manuscript in a comprehensive manner. I believe the revised manuscript could be suitable for publication in Nature Communications.

RESPONSE TO REVIEWERS

Reviewer #1 (Remarks to the Reviewer):

I am satisfied with the authors' responses as they have addressed all my concerns.

We thank the reviewer for their constructive comments and support of our work.

Reviewer #2 (Remarks to the Reviewer):

While some of the critiques have been addressed, I still have reservations about publication of this study given my major critiques have not been adequately addressed.

Conceptual

Responses to Reviewer 2 pts 1, 2,3 and 4:

The caveats raised by all the rebuttal arguments are not sufficient to support the lack of identification of the mechanism. As the authors state correctly bone marrow cell, MSC and resident cardiac cell populations have led to EF improvements of around 7-11%. However, the problem is:

- 1. These responses can be variable lab-lab and without knowing the underlying mechanism its very difficult to troubleshoot why this is the case*
- 2. Translation into the clinic will also require reproducibility, studies in a variety of animal models and stringent quality control of end stage production. As the mechanism is undefined this is a huge barrier to translation and would create a range of difficulties along the way.*
- 3. In this manuscript the improvement in EF is roughly 2 fold higher than in the other cellular studies. It is paramount to discover why this is the case because this could potentially enhance the therapeutic efficacy further.*

Response to Reviewer's points 1-3: We are exploring multiple potential mechanisms for CardioCluster therapeutic efficacy with ongoing studies. The studies described in this report represent a first step in testing and characterization that will inevitably lead to more studies in additional animal models. The abundance of results in this submission supports our initial conclusions, with subsequent studies to define the likely (multiple) mechanisms. Clearly, variability in reported findings between laboratories remains an ongoing challenge for reproducibility in the cell therapy field driven by lack of careful adherence to protocol together with variation in experimental design. Therefore, we are stringent in our study design, inclusion criteria, and documentation of technical details. This careful and rigorous approach and methodological detail should enable other laboratories to reproduce our findings. Regarding a potential mechanism suggested by our data, the manuscript has been re-arranged to highlight transcriptomic profiling in Figure 9 with additional insights now included (pages 23-26). Additional thoughts on potential mechanistic explanations for enhanced therapeutic benefit of CardioClusters are provided below (see *Responses to Reviewer 2, #8*). Lastly, while we are encouraged by the potential translational value, this study is intended to serve as the initial demonstration of CardioCluster engineering and efficacy. Preclinical studies require additional experiments in large animal translational models that would be the subject of future investigation.

- 4. The authors claim that the CardioChimera study is totally distinct. I disagree with this statement and regard both studies as improving the cellular therapy by combining cell types. If the paper had additional insight into the mechanism then I could be persuaded. I do note the studies showing improved survival after oxidative stress in vitro, but there is some issues with this data too (see below).*

Both studies combine distinct cell types to improve cell therapy but use two very different approaches – the 1st being fusion of two varying cells into a single cell for delivery as single cells (CardioChimeras), the 2nd combining multiple cells into a 3D structure with each of the discrete cells still present upon injection (CardioClusters).

The reviewer has not described the issue with the survival after oxidative stress *in vitro* experiment below as stated, therefore we cannot comment upon that concern.

5. *I am not suggesting that the study does not warrant publication, but I do not think is it of the level I generally read at Nature Communications.*

The reviewer is certainly entitled to this subjective opinion, but we have noted other articles in Nature Communications of similar level. For example:

Soon-Jung Park, Ri Youn Kim, Bong-Woo Park, Sunghun Lee, Seong Woo Choi, Jae-Hyun Park, Jong Jin Choi, Seok-Won Kim, Jinah Jang, Dong-Woo Cho, Hyung-Min Chung, Sung-Hwan Moon, Kiwon Ban & Hun-Jun Park. Dual stem cell therapy synergistically improves cardiac function and vascular regeneration following myocardial infarction. *Nature Communications* 10, 3123 (2019).

Major

6. *Responses to Reviewer 2 pt 2: Quoting change in EF from week 1 to week 20 is flawed if the cells are given at time of MI (which the methods suggest). If part of the mechanism is truly a reduction in CM death in the face of oxidative stress, then one would expect a big difference after 1 week as nearly all the CM death following MI will occur within this period. Apoptosis after delivery in vivo would need to be quantified.*

Nearly equivalent EF is present between all groups at Week 1 post-MI, therefore acute reduction in CM death is unlikely to be a major contributor to the mechanism of CardioCluster efficacy. We never asserted reduction in acute cardiomyocyte death as a primary mechanism, especially in view of the longitudinal time course findings that show progressive recovery and improvement over a period of months following delivery. Note that inclusion / exclusion criteria for comparable infarct size were set to insure reproducibility in our study.

7. *Responses to Reviewer 2 pt 2: In the previous study on CardioChimeras there does appear to be a bit of a difference at week 1 which would probably become significant with more animal numbers, followed by either declining function in controls or sustained function in some of the treatments. Given in both the CardioChimera and this CardioCluster study the cells are delivered at the time of MI (as far as the methods state) quoting the delta EF from week 1 to 18 or 20 is flawed. I was quoting the overall magnitude of the effect relative to the controls. If the authors want to study delta EF properly the experimental design needs to be altered to deliver the cells at week 1 (but I understand this would place a big stress on the animals and is the reason it was not designed this way).*

This feedback is somewhat perplexing. We appreciate the concept of delta EF comparability, but standard presentation of results in this field is to calculate magnitude of improvement relative to untreated controls at corresponding longitudinal time point(s). We report values for Week 1 to demonstrate comparability of infarction injury among all groups, which is important for reproducibility as well as demonstration that amelioration of acute damage is not the operative cardioprotective mechanism. Alternatively, Week 18 demonstrates relative improvement for each cell treatment group versus the control population at the same time point, e.g. treated versus no cell therapy, comparable infarction damage at Week 1, ongoing myocardial functional impairment.

The reviewer also asked and answered the question of stress from a technical perspective by stating that delivery of the cell therapy at one week post infarction is not technically feasible. Formation of adhesions between the pericardium and chest wall render mice incapable of withstanding a second surgical intervention. Our experimental protocol of cell delivery at the time of infarction is standard for the field as documented in literally hundreds of published murine

infarction studies. In contrast, a small number of studies with delayed cell delivery involve echo-guided intramyocardial injection – a technique fraught with potential reproducibility concerns for a complex study such as ours. Therefore, we continue to use the standard of the field as previously published in our legacy of work in adoptive transfer cell therapy studies.

8. Responses to Reviewer 2 pt 4: The major increased factor with the CardioCluster seems to be IGF-1 (Supp Fig 8). This gets back to my mechanistic question, how does the efficacy of CardioClusters compare with delivery of say IGF-1 in this example following MI (delivered via eg. mini-pumps or over-expression using AAV mediated delivery to the myocardium)? Approaches such as this would be a much more straightforward to translate.

Previous studies have addressed IGF-1 over-expression and, as the reviewer suggested previously, we have added several of these references to the manuscript including

- Haider, H., Jiang, S., Idris, N.M. & Ashraf, M. IGF-1-overexpressing mesenchymal stem cells accelerate bone marrow stem cell mobilization via paracrine activation of SDF-1alpha/CXCR4 signaling to promote myocardial repair. *Circ Res* 103, 1300-1308 (2008)
- Jackson, R. *et al.* Paracrine Engineering of Human Cardiac Stem Cells With Insulin-Like Growth Factor 1 Enhances Myocardial Repair. *J Am Heart Assoc* 4, e002104 (2015).

However, we do not favor the reductionist concept that a single factor operates as the underlying “mechanism” of why CardioClusters are more efficacious than single cell delivery. Concurrent secretion of multiple factors working synergistically over time is more likely coupled with the persistence of cells delivered as CardioClusters. Furthermore, the panel of factors responsible for observed beneficial effects likely changes over time as the environment within the heart adapts to the injury. To identify each of these factors over the entire 20-week time span is beyond the scope of this current manuscript. We are exploring multiple potential mechanisms for CardioCluster therapeutic efficacy in ongoing studies. In summary, performing an additional 20-week MI study addressing the role of IGF-1 alone is unlikely to resolve the issue of mechanism sought by the reviewer and is beyond the scope of the present study.

Minor (potentially)

9. Review 1 pt2: This raises a concern for me. The cells should also be biologically matched as well. The sequencing should have been run from the same initial 3 cultures (ie thawed at the same time), passaged to make both CC and 2D time matched cultures and then RNA harvested. Can the authors confirm/please add to the methods, their experimental design?

Cells were biologically matched in our analyses as recommended by the reviewer for exactly the reasons provided in the comment. This information has now been added to the single-cell RNA-seq methods section (page 8, lines 5-7): “To ensure samples were biologically matched, the three cardiac cell lines were thawed at the same time and passaged in matched sets for both 3D CardioCluster formation as well as 2D parental cultures.”

Reviewer #3 (Remarks to the Reviewer):

I'd like to thank the authors for taking the time to address the raised questions about this manuscript in a comprehensive manner. I believe the revised manuscript could be suitable for publication in Nature Communications.

We thank the reviewer for their constructive comments and support of our work.

Reviewers' Comments:

Reviewer #3:

Remarks to the Author:

In the present study, the authors use multiple type cells in 3D formation (cluster) for repairing the MI model. The cluster size, cell ratio could be controlled as well. And the most important advantage of this cluster-based technology compared with the previous single-cell method, as far as I am concerned, is its lower cell loss and higher cell retention after injection.

Nonetheless, regarding the EF improvements between the present study and previous study, I would like to say that the author may not be able to conclude that the cardioclusters could lead to a better EF improvement compared with cardiomyocytes. The reason is that the baselines (starting value) of Ejection Fraction ratio for both studies are quite different (~25% for present study vs. ~40% for previous work). This variation could be caused by the experimental variation (person, skill, animal batch, et al.). For example, if the ligation in left anterior descending artery is deeper/wider in the myocardium, the MI area will be larger and the heart will have a lower baseline of Ejection Fraction ratio. I consider that the difference in the baseline could have a dramatic impact on the evaluation of EF improvement. It is more appropriate to compare the repairing effect when the baseline is similar among different groups.

Since all the three cell types within cardiocluster can hardly be differentiated into cardiomyocytes, which are essentially needed for healing the MI area, in the future study the authors are suggested to supplement the stem-cell-derived cardiomyocyte to the cardiocluster for improved therapy development. For example, there are already reports on single cardiomyocyte injection showing long-term high retention [1-3]. The injection of cardiocluster made of cardiomyocytes and other cells may further improve the therapeutic effects.

Reference:

1 Chong JJ, Yang X, Don CW, Minami E, Liu Y-W, Weyers JJ, Mahoney WM, Van Biber B, Cook SM, Palpant NJ (2014) Human embryonic-stem-cell-derived cardiomyocytes regenerate non-human primate hearts. *Nature* 510 (7504):273-277

2 Shiba Y, Gomibuchi T, Seto T, Wada Y, Ichimura H, Tanaka Y, Ogasawara T, Okada K, Shiba N, Sakamoto K (2016) Allogeneic transplantation of iPS cell-derived cardiomyocytes regenerates primate hearts. *Nature* 538 (7625):388-391

3 Liu Y-W, Chen B, Yang X, Fugate JA, Kalucki FA, Futakuchi-Tsuchida A, Couture L, Vogel KW, Astley CA, Baldessari A (2018) Human embryonic stem cell-derived cardiomyocytes restore function in infarcted hearts of non-human primates. *Nature biotechnology* 36 (7):597-605

REVIEWERS' COMMENTS:

Reviewer #3 (Remarks to the Author):

In the present study, the authors use multiple type cells in 3D formation (cluster) for repairing the MI model. The cluster size, cell ratio could be controlled as well. And the most important advantage of this cluster-based technology compared with the previous single-cell method, as far as I am concerned, is its lower cell loss and higher cell retention after injection.

Nonetheless, regarding the EF improvements between the present study and previous study, I would like to say that the author may not be able to conclude that the cardioclusters could lead to a better EF improvement compared with cardiochrmeras. The reason is that the baselines (starting value) of Ejection Fraction ratio for both studies are quite different (~25% for present study vs. ~40% for previous work). This variation could be caused by the experimental variation (person, skill, animal batch, et al.). For example, if the ligation in left anterior descending artery is deeper/wider in the myocardium, the MI area will be larger and the heart will have a lower baseline of Ejection Fraction ratio. I consider that the difference in the baseline could have a dramatic impact on the evaluation of EF improvement. It is more appropriate to compare the repairing effect when the baseline is similar among different groups.

Since all the three cell types within cardiocluster can hardly be differentiated into cardiomyocytes, which are essentially needed for healing the MI area, in the future study the authors are suggested to supplement the stem-cell-derived cardiomyocyte to the cardiocluster for improved therapy development. For example, there are already reports on single cardiomyocyte injection showing long-term high retention [1-3]. The injection of cardiocluster made of cardiomyocytes and other cells may further improve the therapeutic effects.

Reference:

- 1 Chong JJ, Yang X, Don CW, Minami E, Liu Y-W, Weyers JJ, Mahoney WM, Van Biber B, Cook SM, Palpant NJ (2014) Human embryonic-stem-cell-derived cardiomyocytes regenerate non-human primate hearts. *Nature* 510 (7504):273-277
- 2 Shiba Y, Gomibuchi T, Seto T, Wada Y, Ichimura H, Tanaka Y, Ogasawara T, Okada K, Shiba N, Sakamoto K (2016) Allogeneic transplantation of iPS cell-derived cardiomyocytes regenerates primate hearts. *Nature* 538 (7625):388-391
- 3 Liu Y-W, Chen B, Yang X, Fugate JA, Kalucki FA, Futakuchi-Tsuchida A, Couture L, Vogel KW, Astley CA, Baldessari A (2018) Human embryonic stem cell-derived cardiomyocytes restore function in infarcted hearts of non-human primates. *Nature biotechnology* 36 (7):597-605

We thank the reviewer for their constructive comments. We agree that it would be interesting to supplement CardioClusters with stem-cell-derived cardiomyocyte and have added this to the discussion (page 26, line 3-6). "Additional strategies to further enhance the CardioCluster concept could involve incorporating stem-cell derived cardiomyocytes⁹⁰ or even genetically modifying cells with pro-survival factors such as Pim-1⁷⁸ or chemokine receptor (CCR1) to enhance migration, survival and engraftment⁹¹."